# Interplay between Hormones and Several Abiotic Stress Conditions on *Arabidopsis thaliana* Primary Root Development

**DOI:** 10.3390/cells9122576

**Published:** 2020-12-01

**Authors:** Brenda Anabel López-Ruiz, Estephania Zluhan-Martínez, María de la Paz Sánchez, Elena R. Álvarez-Buylla, Adriana Garay-Arroyo

**Affiliations:** 1Laboratorio de Genética Molecular, Desarrollo y Evolución de Plantas, Departamento de Ecología Funcional, Instituto de Ecología, Universidad Nacional Autónoma de Mexico, Mexico City 04510, Mexico; ana_bell_89@hotmail.com (B.A.L.-R.); ezluhanm@gmail.com (E.Z.-M.); mpsanchez@iecologia.unam.mx (M.d.l.P.S.); elenabuylla@protonmail.com (E.R.Á.-B.); 2Centro de Ciencias de la Complejidad, Universidad Nacional Autónoma de Mexico, Mexico City 04510, Mexico

**Keywords:** *Arabidopsis thaliana*, hormones, signaling pathways, primary root development, abiotic stress conditions

## Abstract

As sessile organisms, plants must adjust their growth to withstand several environmental conditions. The root is a crucial organ for plant survival as it is responsible for water and nutrient acquisition from the soil and has high phenotypic plasticity in response to a lack or excess of them. How plants sense and transduce their external conditions to achieve development, is still a matter of investigation and hormones play fundamental roles. Hormones are small molecules essential for plant growth and their function is modulated in response to stress environmental conditions and internal cues to adjust plant development. This review was motivated by the need to explore how *Arabidopsis thaliana* primary root differentially sense and transduce external conditions to modify its development and how hormone-mediated pathways contribute to achieve it. To accomplish this, we discuss available data of primary root growth phenotype under several hormone loss or gain of function mutants or exogenous application of compounds that affect hormone concentration in several abiotic stress conditions. This review shows how different hormones could promote or inhibit primary root development in *A. thaliana* depending on their growth in several environmental conditions. Interestingly, the only hormone that always acts as a promoter of primary root development is gibberellins.

## 1. Introduction

The roots, besides affording structural support to the plant, play crucial roles in the acquisition of nutrients and water as well as in the interaction with different organisms, reacting plastically to diverse stimuli [1]. Therefore, the root system architecture results from the coordinated control of genetic endogenous programs that depends on the plant species and the action of abiotic and biotic environmental stimuli [2]. In this regard, the features of *Arabidopsis thaliana* (hereafter Arabidopsis) primary root (PR) has become an invaluable model to study one of the most relevant processes during plant growth and root plasticity: the homeostasis between proliferation and differentiation that regulate growth under different internal and external stimuli. The Arabidopsis PR has a simple cellular organization that has been greatly described with few cell types concentrically arranged around a central vascular tissue. These cell layers are disposed in the following order: in the outermost layer is the lateral root cap (only present in the meristematic area), followed by the epidermis, the cortex, the endodermis and the pericycle which surrounds the central vascular tissue as the innermost layer [3,4]. Cell proliferation and cell differentiation could be studied in the longitudinal root axis of the PR at any time during early development and this level of cellular and molecular detail is difficult to find in any other organ [1,5,6]. The PR pattern is established during embryogenesis and its root apical meristem (RAM) participates post-embryologically in the development that determines the final root architecture [1,4]. A longitudinal section through the PR shows the distinct developmental zones that are responsible for the final size of the PR; from the root tip upward, is the meristematic zone (MZ) followed by the elongation zone (EZ) and, lastly the differentiation zone (DZ) [7]. The meristem is composed of the stem cell niche (SCN) and two domains: the proliferation domain (PD) and the transition domain (TD) [8,9]. The SCN includes a central organizer (quiescent center, QC) enclosed by four different initial cells that divide asymmetrically to maintain themselves and to provide cells that will divide symmetrically to form the PD in the basal part of the root and to provide cells to form the columella in the apical part of the root; the columella and the columella root cap have a protective and transmitter role of environmental conditions [4,10,11,12]. The meristematic zone has a mitotically active cell population derived from the SCN, whereas cell elongation occurs in the EZ and cell differentiation in the DZ [7]. As the root morphology and cellular features depend on a coordinate interplay between intrinsic and extrinsic signals, this review was motivated by the need to explore how Arabidopsis PR sense and transduce their external conditions to modify its development and how hormone-mediated pathways contribute to achieving this goal. While hormone gene regulation is fundamental to root development, we specifically focused on phenotypic evidence regarding PR growth under either exogenous application of compounds that change the hormone concentration and examples of loss or gain of function mutants on key components of hormone pathways. To this end, we first briefly delineate the hormone function in PR growth under control conditions and then describe the different functional role of each phytohormones alone or in crosstalk with other hormones during Arabidopsis PR growth under different abiotic stress conditions, such as low water availability (osmotic, salt and cold), oxidative stress, metal stress, high temperature stress, soil alkalinity and nutrient condition. To our knowledge, this is the first review that concentrates on the Arabidopsis PR growth under so many different abiotic stress conditions. This comparative analysis shows how the same hormone could either promote or repress PR growth depending on the stress. Interestingly we have found with this approximation that GA is the only hormone that promotes primary root growth in all stress conditions described in this review.

## 2. Hormone Participation in Arabidopsis Primary Root Development

All phytohormone functions depend on the interplay between biosynthesis, catabolism, signaling pathway, conjugation and transport to regulate different PR cell processes like proliferation, quiescence, elongation and differentiation. PR growth is a rapid and sensitive parameter to evaluate, in vivo, the interplay between hormones and the impact of different stress conditions on development. So far, the major hormones studied until now are auxins and cytokinins (CK) followed by brassinosteroids (BR) gibberellins (GA), abscisic acid (ABA), ethylene, salicylic acid (SA), jasmonates (JA) and strigolactones (SL). As many previous reviews have described biosynthesis, catabolism, signaling, and transport of these hormones [13,14,15,16,17,18,19,20,21,22,23,24,25], we briefly describe the generalities of hormone function in root development and summarize the function and names of the proteins encoded by genes, whose loss and gain-of-function (LoF and GoF) mutants were analysed in this review (Appendix A).

Auxin: Auxin is fundamental for root patterning and growth [14]. In PR growth, auxin concentration forms a gradient, being its maximum accumulation at the root apex, specifically in the QC and with a gradual decrease towards the meristematic zone and the EZ. The accumulation in the QC is firstly generated by the directional and organized transport of auxins from the shoot to the root and later by root auxin synthesis [9,26,27,28,29]. Auxin can move cell to cell via polar auxin transport using auxin influx carriers AUXIN1 (AUX1)/LIKE-AUXs (LAXs) and auxin efflux transporters PIN-FORMED family (PINs) and ABCB/MULTIDRUG RESISTANCE (MDR)/PHOSPHOGLYCOPROTEIN (PGPs) protein family [30,31]. Interestingly, PIN genes have been deeply studied as they code for transmembrane proteins that are asymmetrically localized either in the plasma membrane or in the endoplasmic reticulum to facilitate auxin polarly movement affecting PR growth [32,33]. Although auxin has a biphasic impact in PR growth where low concentration promotes it while high concentrations inhibit it (Figure 1A) [34]; loss of function mutants of auxin transport, signaling and biosynthesis, have shorter PR growth showing its promoter activity in PR development (Figure 1A) [35,36,37,38,39,40,41].

CK: There is a CK gradient in the PR with a maximum in the lateral root cap, columella and the SCN [42]. CKs are commonly described as negative regulators of PR growth, as exogenous CK application generates short PRs with shorter meristems (Figure 1A) [43]. Accordingly, the plants overexpressing (OE) genes that function positively in the CK biosynthesis, have shorter roots, while mutants of genes that lower CK concentration (in CK signaling pathway or biosynthesis) have longer PRs than WT plants (Figure 1A) [43,44,45,46,47]. Interestingly, CK receptors play different roles in PR growth; it has been described that the mutant in the *ARABIDOPSIS HISTIDINE-KINASE 3* receptor, *ahk3-3* has longer PRs; while the triple mutant *ahk2-1 ahk3-1 ahk4-1* has a drastic reduction in PR growth and the double mutant *ahk2-1 ahk3-1* and the single mutants *(ahk2-1, ahk3-1, and ahk4-1)* do not have changes in the PR length compared to WT plants [43,48,49].

Auxin-CK crosstalk: The CK/auxin ratio partially establishes PR zonation: higher concentration of CK with respect to auxin in the TD allows the cells to transit from the MZ to the DZ due to the following: CK regulates the type B CK *ARABIDOPSIS RESPONSE REGULATOR 1 (ARR1)* expression whose protein, together with ARR12, another type-B ARR, activates *SHORT HYPOCOTYL2 (SHY2/IAA3)* expression. SHY2 protein acts as a transcriptional repressor of several PIN genes, thus altering auxin transport and lowering the auxin concentration in the TD [50,51,52,53].

GA: GA positively regulates PR growth affecting cell proliferation and cell differentiation. The exogenous application of the bioactive GA (GA3) promotes PR growth [54]. Moreover, the hormone concentration is positively correlated with cell length: high levels are found in the EZ while low levels are present in the MZ [55,56]. Interestingly, GA synthesized in the PRs accumulates in the endodermis of the EZ to regulate root growth [21,57]. Finally, the reduction of GA levels either by the inhibition of GA synthesis, the loss of function of GA biosynthetic genes or the accumulation of DELLAs proteins result in reduced PR growth (Figure 1A) and reduced root meristem [57,58,59]. DELLAs proteins are negative regulators of the GA signaling pathway that function as inhibitors of plant development and their effects are reverted by GA [60,61].

BR: BR has a biphasic behaviour regarding PR development as auxins; high concentrations can inhibit PR growth and low concentrations promote it (Figure 1A) [20]. However, according to the short PR phenotype of the BR insensitive mutant of the BR receptor (*BRI1*), *brassinosteroid insensitive 1-116* (*bri1-116*), BR functions as a promoter of PR growth (Figure 1A) [20,62,63]. BR controls both root meristem size and cell elongation through its effect in the epidermis cell type [63,64,65].

ABA: ABA is another hormone that also regulates PR growth in a concentration-dependent manner, phytohormone exogenously applied in low concentrations stimulates, while at high concentrations inhibits PR growth (Figure 1A) [66,67]. As CK, ABA participates in RAM premature differentiation [68,69] and the loss of function mutants of *aba1*, with reduced levels of ABA production due to a mutation in the *ZEAXANTHIN EPOXIDASE* gene that is required for ABA synthesis, have shorter PRs than WT plants showing, as with auxin, its promoter function (Figure 1A) [70].

Ethylene: Exogenous ethylene addition negatively regulates PR growth (Figure 1A) affecting cell proliferation and cell elongation [71,72,73]. In the PD this phytohormone increases the levels of the *INHIBITOR/INTERACTOR OF CDK 1/KIP-RELATED PROTEIN 1* (*ICK1/KRP1*), an inhibitor of the cell cycle; while in the EZ, ethylene inhibits cell elongation mainly through cross-talk with auxins via the epidermis tissue [72,73,74]. In addition, the loss of function mutants in the *ETHYLENE INSENSITIVE* (*EIN2)* gene, that encodes a membrane-bound protein crucial for the ethylene signaling, and in the ethylene receptor *ETHYLENE RESPONSE* (*ETR1*), have longer PRs. Accordingly, the mutant of the *ETHYLENE OVERPRODUCER 1* gene (*ETO1*) has a shorter root than WT plants (Figure 1A) [75]. *eto1* mutants have high ethylene levels due to a truncated ETO1 protein incapable of target for proteosomal degradation to the 1-AMINOCYCLOPROPANE-1-CARBOXYLATE (ACC) SYNTHASE (ACS) protein, required for ethylene synthesis [76].

Auxin-ethylene crosstalk: As we described above, ethylene negatively regulates PR growth while low levels of auxin have a promoter effect. However, ethylene treatment promotes auxin accumulation in the EZ of the PR to inhibit root cell elongation [72,77]. Consistently, the PR growth of the auxin receptor mutant *transport inhibitor response 1* (*tir1*), the *auxin transporter protein 1* (*aux1-7*) and the gain of function mutant *indole-3-acetic acid inducible 17*/*auxin resistant 3* (*iaa17*/*axr3-1*), a mutant in the repressor of auxin-inducible gene expression, is less sensitivity to ethylene [72,77].

JA: JA negatively regulates PR growth [78,79]. JA exogenously applied results in plants with shorter PRs, with a reduced meristem size, shorter fully elongated cells and less meristem cell number than WT plants (Figure 1A) [22,78,80]. Furthermore, loss of function mutants of *NOVEL INTERACTOR OF JAZ* (*NINJA*), that negatively regulates JA signaling, have shorter PRs than WT plants, due to a defect in cell elongation (Figure 1A) [80].

SA: SA is a negative regulator of PR growth in a concentration-dependent manner (Figure 1A) [81,82,83]. Moreover, SA attenuates PR growth through crosstalk with PIN-dependent auxin distribution [82,84].

SL: SL, like auxin, ABA and BR, is a hormone that also regulates PR growth in a concentration fashion manner. Low concentrations stimulate PR growth and high concentrations inhibit it (Figure 1A) [85]. Despite the phenotype obtained with the phytohormone exogenously applied, the loss of function of the F-box protein MORE AXILLARY GROWTH 2 (*max2*), which participates in SL signaling and mutations in biosynthesis genes: *CYTOCHROME P450 MONOOXYGENASE* (*max1*), *CAROTENOID CLEAVAGE DIOXYGENASE 7* (*max3*) and *CAROTENOID CLEAVAGE DIOXYGENASE 8* (*max4*) have shorter PR than WT plants emphasizing its function as growth promoter (Figure 1A) [85]. As SA, SL down regulates auxin concentration and down regulates the expression of the auxin efflux carriers *PIN2*, *3* and *PIN7* [85].

SL and CK: There is a complex interaction between SL and CK regarding PR development: *max2-1* has a shorter whereas the double *ahk2-5 ahk3-7* has longer PR and the triple mutant *ahk2-5 ahk3-7 max2* has the longest PR growth compared to the single and double mutants, and to the WT [86].

All of these data show that the function of hormones on PR growth under controlled conditions is regulated as follows: the exogenous application of GA stimulates PR growth whereas the application of CK, SA, ethylene and JA suppress it. In addition, auxin, SL, ABA and BR have a biphasic behaviour that depends on their concentration; high concentration inhibits whereas low concentration stimulates PR growth (Figure 1A). Interestingly, the study of PR growth in mutants of the hormones with a biphasic behaviour and GA, show that these hormones are promoters of growth and have short PRs, whereas JA and ethylene mutants show that they are negative regulators of PR growth. Moreover, CK synthesis and signaling act as negative regulators, while CK receptors can play different roles in PR growth. Despite all this information, it is important to consider that PR growth phenotypes depend on the spatial endogenous concentration of the phytohormones, their crosstalk and the impact of other internal and environmental stress conditions on this organ development (Figure 1B).

## 3. The Role of Different Hormones on Primary Root Growth under Several Abiotic Stress Conditions

Plant plasticity is the capacity of a single genotype to dynamically adjust its development to diverse internal and environmental conditions [87]. Phytohormones are one of the transducers of these internal and external conditions that allow survival and plasticity of plants [88,89]. In the next sections we will summarize the current information on the role of several phytohormones alone or in conjunction, as transducers of specific environmental conditions and how they differentially shape Arabidopsis PR development (Figure 1B).

### 3.1. Low Water Availability

Abiotic stresses such as drought, salinity, osmotic and cold stress affect water availability, plant development and productivity [90,91]. Because the root is the first organ to sense water deficiency, PR growth and, in general, the root system architecture, is highly modulated by these different stresses [92,93]. In Arabidopsis, the PR length is reduced in response to water deficiency although there is an increment in the root/shoot ratio [94]. Also, it has been shown that low and high osmotic and salt stress conditions induce and repress PR growth, respectively [95,96]. The salt stress is produced by the excessive uptake of sodium (Na+) and chloride (Cl-) ions, that not only can disrupt the osmotic balance but also have a cytotoxic effect [97,98]. Osmotic stress, on the other hand, is originated by changes in the solute concentration outside the cells, modifying the movement of water across the cell membrane [99]. The compounds used to stimulate osmotic stress conditions are the non-ionic compounds, such as Polyethylene-glycol (PEG), mannitol and sorbitol and also, the ionic compound, NaCl, [100,101,102]. Under salt stress conditions, the PR initial growth reduction is due to the osmotic pressure of the salt and later is caused by the inability of the roots to contend with the toxic effect of the ions [103]. Under both osmotic and salt stress conditions, the meristem cell number and the length of the completely elongated cells are affected [95,104,105]. In addition, many studies have shown that ABA concentration increases during drought stress and helps to withstand PR growth under low water availability conditions [95,105,106,107,108]. Finally, cold stress is produced when plants are exposed to temperatures ranging from 0 to 15 °C, under these conditions water availability is also reduced and PR growth is inhibited [109,110,111]. 

#### 3.1.1. Phytohormone Roles in Primary Root Growth under Osmotic or Salt Stress Conditions

ABA: Endogenous ABA concentrations increased drought and salt tolerance in Arabidopsis and can function as an indicator of soil water availability [68,106,112]. This phytohormone is synthesized and accumulated in roots under low water availability and functions in a concentration-dependent manner (Figure 1A) [95,113,114,115]. ABA is necessary for controlling root growth under drought, osmotic and salt stresses and root growth is more affected when the ABA signaling pathway is disturbed mainly in the endodermis but also in the pericycle [116,117,118]. Moreover, ABA-dependent pathways regulate the expression of the cold responsive (COR) genes, which are important to withstand cold stress conditions [107,119]. Besides, ABA inhibition by fluridone, an ABA synthesis inhibitor, generates plants with longer PRs under moderate osmotic stress conditions (−0.37 MPa) compared to WT plants (Figure 2A) [95]. Interestingly, fluridone treatment prevents relative PR growth rate under high salt stress conditions (140 mM) both, in the recovery and in the homeostasis phases (Figure 2A) [118]. These contrasting results could be either that ABA has a dual function, depending on the type of stress imposed or that the different response depends on the methodological procedures (e.g., the osmotic potential of the stress condition).

Auxin: The accumulation of this hormone decreases under osmotic stress conditions as reported by the lower activity of the reporter driven by the auxin response elements (AuxRE), referred to as *DR5,* and higher levels of DII-Venus an AUXIN/INDOLE ACETIC ACID (Aux/IAA)-based auxin signaling sensor [120]. Moreover, salt stress also reduces IAA concentration, changes auxin distribution and decreases the activity of the *DR5* auxin reporter [93,121]. In addition, PIN proteins are continuously cycling between clathrin-mediated endocytosis and exocytosis and hyperosmotic stress enhances endocytosis while hypoosmotic stress increases exocytosis altering the equilibrium between endo and exocytosis in root meristem cells, thus changing auxin distribution and probably, root patterning [122]. Curiously, the PR of plants under osmotic and salt stress conditions are shorter in auxin transport mutants and longer in auxin signaling mutants (Figure 2A and Appendix A). For instance, *aux1-7*, an auxin influx transporter mutant, has a shorter PR than WT plants under osmotic and salt stress conditions [115,123]. Besides, *pin2*, an auxin efflux transport mutant has a short PR, while the double mutant of auxin receptors (*tir1* with the mutant *auxin signaling f-box protein* 2 (*afb2*)) has longer PR than WT plants under salt stress [121,124]. In addition, the *iaa-ala hydrolase 3* (*iar3*) loss of function mutant, a hydrolase that releases active IAA from inactive IAA storage, has larger PRs in response to osmotic stress (Appendix A) [125]. Interestingly, under moderate stress conditions (−0.37 MPa), PR phenotype can be rescued by auxin addition [95].

CK: Drought decreases the production of CKs via repression of *ISOPENTENYL TRANSFERASE* (*IPT*) genes, responsible for the rate limiting step of CK biosynthesis; and, consequently, enhances PR growth [126,127,128,129]. CK inhibits PR growth under osmotic and salt stress conditions as shown by the OE of *IPT8*; accordingly, the OE of the *CYTOKININ OXIDASE* (*CKX*), which participate in CK catabolism, and the loss of function of *IPT* genes increases the PR growth during salt stress [127,129,130]. The controversy of whether CK plays a negative or a positive function in plant stress response as reported [131] could also be applied to the function of the CK receptors in PR growth under different osmotic potential treatments (Figure 2A). Under moderate stress conditions (–0.7 MPa), the PR growth of *ahk3-3* and *ahk2-2 ahk3-3* are longer, while under 75 mM of salt stress are shorter than WT plants; also the PR growth of the single mutant *ahk2-2* and the double mutants of the CK receptors, *AHKs* including *CYTOKININ RESPONSE 1* (*CRE1/AHK4*)*: ahk2-2 ahk3-3*, *ahk2-2 cre1-12* and *ahk3-3 cre1-12*, is inhibited under low water potential (–1.2 MPa) or 75mM salt stress conditions [112] (Figure 2A and Appendix A). Interestingly, AHK 2, 3 and 4 could act as osmosensors during PR growth under stress conditions, as it has been reported by complementation experiments in yeast [112].

GA: Salt stress reduces GA levels and, consequently, enhances DELLAs protein accumulation, that act as negative regulators of GA signaling [117]. DELLAs proteins are inhibitors of PR growth under salt and cold stress conditions. The PR growth of the double (*ga insensitive-t6* (*gai-t6*) with *repressor of gai-24* (*rga-24*)) and quadruple (*gai-t6, rga-t2* with *rga-like 1-1* (*rgl1-1*) and *rgl2-1*) DELLAs mutants are longer under salt and cold stress than WT plants (Figure 2A and Appendix A) [117,132]. Also, GA is necessary to recover PR growth from late phases of salt stress conditions [118].

BR: BR participates in the trade-off between growth and stress response [133]. Specifically, BRs are required to withstand osmotic stress conditions and, as with GA, are required for PR growth recovery from late phases of salt stress conditions [118]. The loss of function of the BR receptor *BRI1* delays relative PR growth rate, while the gain of function mutant (*bzr1-D*), in which the transcriptional repressor *BRASSINAZOLE RESISTANT 1* (*BZR1*) is constitutively active, promoted relative PR growth rate under salt stress (Figure 2A) [118]. Accordingly, *bri1* and the triple mutant of *BRI1* with the other BR receptors *BRASSINOSTEROID*
*INSENSITIVE1-LIKE RECEPTOR KINASE* (*BRL1*) and *BRL3* (*bri1 brl1 brl3*) have shorter PRs under osmotic stress conditions than WT plants [134] (Figure 2A and Appendix A).

Ethylene: Ethylene is required to withstand osmotic and salt stress conditions. Treatment of WT plants with NaCl and mannitol inhibit *EIN2* expression, a gene whose protein is involved in the positive regulation of ethylene signaling [135]. In fact, the PR of *ein2-5* is shorter under osmotic and salt stress conditions than WT plants [135]. Accordingly, the PR growth of the OE of the transcription factor (TF) *ETHYLENE RESPONSE FACTOR1* (*ERF1*), that participates in the ethylene signaling pathway, is longer than WT plants under salt stress (Figure 2A and Appendix A) [136].

JA: JA is also a plant hormone that participates in the trade-off between growth and biotic stress response. Interestingly, salt stress conditions activate the transcript accumulation of JA-responsive genes especially in the MZ and in the EZ of the PR, mainly in the stele tissue for *JASMONATE ZIM DOMAIN PROTEIN 5* (*JAZ5*) and in cortical cells for *JAZ7* and *JAZ8*, which act as negative repressors of JA signaling. Thus, the up-regulation of these genes functions to turn-off JA signaling as a mechanism to improve root growth recovery in response to salinity [137]. The PR length of the loss of function mutants of the *CORONATINE INSENSITIVE 1* (*COI1*) gene, that code an F-box protein that forms part of the E3 ubiquitin ligase SCF^COI1^ complex (*coi1-2*); the *MYC2/3/4* which encode bHLH TFs that regulate the expression of JA-responsive genes (*myc2/3/4*)*;* and *JAZ3* (*jai3-1*) that encodes a protein resistant to degradation, are significantly larger than those of the WT under salt treatment (Figure 2A and Appendix A) [118,137].

In summary, auxin signaling, transport and conjugation are required to regulate PR development under osmotic and salt stress conditions, some mutants inhibit whereas others promote PR growth. CK synthesis is necessary to inhibit PR growth under osmotic and salt stress conditions whereas CK catabolism is required under salt stress conditions (Figure 2A and Table 1). Interestingly, single and double mutants in CK receptors affect PR growth in different ways depending on different external osmotic potential showing the promoter and inhibitor role of CK on PR growth under osmotic stress conditions (Figure 2A and Table 1). In addition, under salt stress conditions, the PR growth is always inhibited in the different CK receptors mutants, conferring to CK a promoter role of PR response to this stress (Appendix A). BR and ethylene signaling participate as activators in PR growth under salt and osmotic stress conditions (Figure 2A and Table 1). Furthermore, GA signaling is necessary for PR growth under salt and cold stress conditions, whereas JA signaling functions as a PR growth repressor under salt stress conditions (Figure 2A and Table 1). Finally, ABA is a hormone necessary to control PR growth under low water availability and the inhibition of ABA generates shorter PRs under high salt stress and longer PRs under moderate osmotic stress (Figure 2A and Table 1 and Appendix A).

#### 3.1.2. Interaction between Different Phytohormones and Exogenous ABA Application and Its Function on Primary Root Growth

ABA is a plant hormone necessary to withstand low water availability stress responses and ABA accumulation is necessary for PR maintenance under this stress condition [138]. Regarding mutants, the gain of function mutant of the phosphatase *ABA INSENSITIVE 1* (*ABI1*) whose protein function as a negative regulator of ABA signaling, have longer PRs in response to ABA exogenous application [75,117]. Accordingly, Bhaskara and collaborators [139] found that the loss of function of phosphatase *ABI1*, *HIGHLY ABA-INDUCED 1-2* (*HAI1*) and *AKT1-INTERACTING PHOSPHATASE 1,* that are negative regulators in ABA signaling, in the single (*abi1td* and *abi2td*), doubles (*hai1-2 aip1-1*) and (*hai1-2 hai3-1*) and triple (*hai1-2 aip1-1 hai3-1*) loss of function mutants have shorter PRs under different ABA concentrations. Besides, ABA acts together with other hormones to generate different stress responses; for instance, low water availability increases ABA concentration that inhibits ethylene production and affects auxin distribution in PRs [95]. To understand the ABA crosstalk with other hormones and its relevance on PR growth, we discuss several experiments that have been carried out using ABA exogenous additions.

Auxin-ABA: Low concentration of ABA stimulates PR growth partially by increasing auxin transport and signaling; in addition, high ABA concentration decreases the expression of several genes that encode auxin influx and efflux transport proteins [67,140,141]. Furthermore, auxin influx and efflux transporters are important for the inhibitory and stimulatory effect of high and low ABA concentration on PR growth, respectively, *pin2* is less sensitive to the stimulatory effect of low ABA concentration (1 µM) while *aux1-T* is insensitive to both low and high ABA concentrations (10 µM) [66]. Moreover, ABA affects auxin signaling and distribution and the PRs of the loss of the function of auxin receptor and transporters: *tir1-1*, *aux1-7*, *pin2*, and the gain of function mutant *AXR2/IAA7* (*axr2-2*), are more resistant to high ABA exogenous application than WT plants (Figure 2B) [75,142]. In addition, the loss of function mutant of *AUXIN RESPONSE FACTOR 2* (*ARF2*) (*arf2-101*), an auxin transcription repressor that also acts as a negative regulator of ABA signaling, is hypersensitive to ABA and has shorter PRs, whereas the *ARF2* OE has longer PRs than the mutant *arf2-101*, under ABA exogenous applications [143]. Furthermore, the OE of the auxin biosynthesis gene *YUCCA 4* (*YUC4*) has shorter PRs than WT plants under ABA exogenous application [144] (Figure 2A and Appendix A).

ABA-ethylene: The inhibitory effect of ABA depends on ethylene biosynthesis and signaling [67,145]. Accordingly, loss of function mutants of the ethylene signaling pathway: the receptor *ETHYLENE RESPONSE 1* (*etr1-1*), the positive regulator of ethylene response *ein2-5*, and the sextuples (*acs1–1 acs2–1 acs4–1 acs5–2 acs6–1 acs7–1* and *acs2–1 acs4–1 acs5–2 acs6–1 acs7–1 acs9–1*) and septuple (*acs1–1 acs2–1 acs4–1 acs5–2 acs6–1 acs7–1 acs9–1*) loss of function mutants of ACS enzyme that synthesizes ethylene, have longer PR length than WT plants under ABA exogenous application (Figure 2B and Appendix A) [66,146]. In addition, the overproducer ethylene mutant *eto1-1* has shorter PR under this condition. Interestingly, the effect of ABA on *etr1-1*, *ein2-5* and *eto1-1* is reverted with the co-treatment of ABA with the inhibitor of ethylene biosynthesis aminoethoxyvinylglycine (AVG). Also, the triple recessive mutant of the loss of function of *ABI*s and *HYPERSENSITIVE TO ABA1* (*HAB1*), that are negative regulators of ABA signaling, (*abi1 abi3 hab1*) and whose mutations are hypersensitive to ABA PR inhibition, has longer roots under ABA co-treatment with either AVG or the application of an antagonist of ethylene signaling Ag+ (AgNO3) (Appendix A) [66,146].

Auxin-ethylene-ABA: Inhibitors of either ethylene signaling or biosynthesis and auxin influx block the PR growth inhibition generated with high ABA concentration [66,146]. In addition, auxin and ethylene are required and participate in the same pathway regarding ABA PR growth inhibition as shown by the double mutants phenotypes of *ein2 tir1* and *ein2 axr1*, that are not inhibited by ABA as the single mutants, and have longer PR in response to ABA [75]. Interestingly, the PR growth of the double mutant *ein2 aux1* is less inhibited under ABA application compared to either of the single mutants, suggesting that auxin and ethylene can also contribute alone to ABA responsiveness (Figure 2B) [75].

ABA-CK: It has been postulated that ABA and CK have opposite functions during different growth processes including plant adaptation to several stress conditions. Interestingly, the ABA content is low in mutants with low levels of CK (as seen in the quadruple mutant *ipt1 3 5 7* and in the OE of three different *CKX*) [127]. Accordingly, the loss of function mutant of the cytokinin glycosyltransferase *UGT76C2* (*ugt76c2*), that lowers the levels of active CK, and the OE line of this gene, have longer and shorter PR, respectively, under ABA treatment [147]. Moreover, the PR growth of the double mutants *arr1 arr11*, *arr1 arr12*, *ahk2-2 ahk3-3*, the triple mutant *arr1 arr11 arr12* and the OE of *ARR5*, a type-A ARR gene, are shorter under exogenous ABA applications than WT plants (Figure 2B and Appendix A) [148,149]. In addition, the PR inhibition of the triple mutant *arr1 arr11 arr12* can completely and the OE of *ARR5* can partially be reversed with the double mutant of the *SUCROSE NON-FERMENTING 1-RELATED PROTEIN KINASES 2* (*SNRK2*) genes, *snrk2.2 snrk2.3*, showing that both hormones can function together or alone as PR growth repressors [148].

ABA-GA: ABA stabilizes DELLAs proteins by reducing GA accumulation (Figure 2B) [150]. Moreover, the PR growth of the quadruple DELLAs Arabidopsis mutant (*gai-t6 rga-t2 rgl1-1 rgl2-1*) is less affected than WT plants when exogenous ABA is added (Figure 2B) [117].

ABA-BR: BRASSINOSTEROID INSENSITIVE 2 (BIN2), a negative regulator of BR signaling, also participates in ABA signaling, phosphorylating SnRK2.2 and SnRK2.3, and enhancing its kinase activity [151]. In addition, the PR growth of the triple loss of function mutant of *BIN2* and its two closest paralogs, *BIN2 like1* and *BIN2 like 2* (*BIL1* and *BIL2*)*: bin2-3 bil1 bil2* is less sensitive to ABA PR growth inhibition, while the PR of the gain-of-function mutation of *BIN2* (*bin2-1*) and the loss of function of *BRI1* (*bri1-9*) that are insensitive to BR, are hypersensitive to ABA exogenous application compared to WT plants (Figure 2B and Appendix A) [151,152].

Overall, these data reveal the importance of hormone interactions on the effects induced by ABA-exogenous addition. In brief, the auxin influx transport is important for the inhibitory effect of ABA in PR growth at high and low ABA concentrations, while the auxin efflux transport is required for the stimulatory growth of PR under low ABA concentration (Figure 2B). In addition, the PR is shorter under the co-treatment of ABA with high auxin levels. Also, auxin and ethylene are required and participate together (but also individually) with ABA in PR growth inhibition (Figure 2B). Besides, ABA addition in the presence of GA or BR, promotes PR growth (Figure 2B and Table 1). Moreover, the shorter PR growth observed at low CK levels, can partially be reversed in *snrk2.2 snrk2.3* double mutant background, suggesting that ABA is downstream of CK in this stress condition.

Finally, the function of auxin and CK is similar and the function of BR is the same, in PR growth independently if they are growing under osmotic, salt stress or ABA exogenous application. Furthermore, the function of GA in PR growth is the same in salt cold stress and ABA exogenous application. Interestingly, although ethylene is a hormone that promotes PR growth under osmotic and salt stress conditions, it functions as a PR growth repressors with ABA, suggesting a double ethylene activity that is dependent and independent of ABA in response to low water availability (Figure 2A,B and Table 1).

### 3.2. Primary Root Growth Response under Oxidative Stress Conditions

Aerobic organisms utilize oxygen as a molecular acceptor and generate many reactive oxygen species (ROS) such as superoxide (O_2_^−^) and hydrogen peroxide (H_2_O_2_) that are signaling components produced by metabolic pathways [153,154]. Initially, ROS were considered only by-products of metabolism that at high concentration could be harmful as they oxidize proteins, lipids, nucleic acids and disturb membrane functions. Plants have evolved enzymatic and non-enzymatic ROS detoxifying strategies as a means of minimizing the deleterious effects associated with ROS [155,156]. Recently, these compounds have also been described as important components of many signaling pathways [155]. Interestingly, these compounds are increased to high levels under abiotic and biotic stress conditions and affect profoundly root growth.

The exogenous application of H_2_O_2_ inhibits PR growth through the downregulation of genes involved in the cell cycle [157]. Moreover, in the longitudinal axis of the PR, O_2_^−^ is accumulated in the MZ, where it is required to sustain cellular proliferation, while in the EZ, H_2_O_2_ helps to maintain the cell differentiation (Figure 2C) [158]. Specifically, the mutant of the bHLH TF *UPBEAT1* (*UPB1*), a negative regulator of the expression of a set of peroxidases, alters this ROS balance affecting PR growth, while the OE line has larger PRs than WT plants [158]. In addition, single and double loss of function of a NADPH oxidase that produces ROS, the *ARABIDOPSIS THALIANA RESPIRATORY BURST OXIDASE HOMOLOG F/D* (*atrbohF* and *atrbohD/F*) have shorter PRs than WT plants [159]. Likewise, the PR length of the mutant deficient in glutathione (GSH) synthesis capacity, *root meristemless 1-1* (*rml1-1*), is significantly shorter than WT and similar results are obtained using the GSH-synthesis inhibitor, buthionine sulfoximine (BSO) in WT roots [160].

The phytohormones also participate in ROS response as mediators of its role as signal molecules and/or in detoxifying to maintain normal ROS levels [155,156]

Auxin: The exogenous H_2_O_2_ treatment inhibits PR altering auxin distribution and the PIN proteins localization that becomes uniformly distributed in the cell membrane of root tip cells [161,162]. Moreover, the redox control, maintained by NADPH-dependent thioredoxin reductase (NTR/TRX) and glutathione (GSH/GRX) systems, can affect auxin signaling and transport [163,164]. NTR catalyse disulphide reduction, whereas CADMIUM SENSITIVE (CAD2) is implicated in GSH biosynthesis and both are involved in cellular redox homeostasis [164]. The PR of the triple loss of function mutant of NADPH-dependent thioredoxin reductases a/b and *CAD2*: *ntra ntrb cad2*, is shorter than WT plants and auxin exogenous application in low concentrations partially restores PR growth inhibition of this mutant [164]. In addition, under Methyl Viologen (MV), a compound that induces ROS generation, the double mutants of auxin receptors, *tir1 afb2* and *tir1 afb3*, have longer PRs than WT plants (Figure 2C and Appendix A) [124]. Furthermore, *tir1 afb2* mutant has lower H_2_O_2_ levels compared to WT in response to MV, and interestingly, under control conditions, this double mutant positively regulates the activity of the antioxidant enzyme ASCORBATE PEROXIDASE 1 (APX1) [124].

GA: DELLAs cause low ROS levels, promoting the expression of antioxidant enzyme genes and restraining PR growth under abiotic stress in order to increase plant survival [165]. The GA deficient loss of function mutant *ga1-3*, which is a mutant of the *ENT-COPALYL DIPHOSPHATE SYNTHASE* (*GA1/CPS*) gene, that participates in GA synthesis and accumulates more DELLAs proteins, has lower ROS levels and a shorter PR than WT plants. Additionally, *ga1-3* is less inhibited with the exogenous application of an inhibitor of ROS production, Diphenylene iodonium (DPI). This phenotype is reverted in the quintuple loss of function mutant of *ga1-3* with four DELLAs genes (*ga1-3 gai-t6 rga-t2 rgl1-1 rgl2-1*), which exhibits higher levels of ROS in roots than *ga1-3* single mutant, and a PR length like WT but more sensitive to DPI root inhibition (Figure 2C and Appendix A) [165].

BR: In PRs, the level of H_2_O_2_ increases whereas the level of O_2_^−^ decreases under BR treatment [166,167]. The mutant of a steroid 5α-reductase (*deetiolated 2* (*det2-9*)) that acts in the BR synthesis pathway, has hyperaccumulation of O_2_^−^ and shorter PR growth. Accordingly, *det2-9* has longer PRs in the presence of superoxide dismutase (SOD) or in the addition of 1,3-dimethyl-2-thiourea (DMTU), two oxygen radical scavengers, than in control plants (Figure 2C and Appendix A) [166].

ABA: ABA treatment (30 µM) induces H_2_O_2_ production at the root tip and the co-treatment of ABA with ROS scavenging reagent GSH or Dithiothreitol (DTT), partially releases the PR inhibition induced by this hormone [168,169]. Furthermore, the single loss of function *atrbohF* and the double mutant (*atrboh D/F*) of NADPH oxidases that produce ROS, have a PR growth insensitive to ABA exogenous application (10 µM) and have reduced ROS production in roots compared to WT plants [159,169]. Accordingly, under a higher concentration of ABA (50 µM), the PR of the single mutants *atrbohD1*, *atrbohF1* and *atrbohC*, and the double mutants *atrbohD1/F1* and *atrbohD2/F2* are less inhibited under ABA treatment suggesting that ABA-dependent PR growth inhibition requires H_2_O_2_ as a second messenger. (Figure 2C) [169,170].

Ethylene: In PRs, the level of O_2_^−^ increases under ACC treatment and decreases under AVG treatment, that inhibits ethylene biosynthesis [166]. Moreover, the enzymatic activity of enzymes like peroxidases increases whereas catalases decrease in the *ein2-5* mutant [171]. Despite the increment of O_2_^−^ with ethylene, the PR growth of *ein2-5* and the double mutant of *ein3* with *ein3-like 1* (*eil1-1*), are longer under MV exogenous application than WT plants (Figure 2C and Appendix A) [166].

#### Hormone Crosstalk under Oxidative Stress Conditions

Auxin-ABA: The exogenous application of the N-1-naphthylphthalamic acid (NPA), an auxin efflux transport inhibitor, increases the ABA inhibitory effects in the PR growth of the NADPH oxidase mutants (*atrbohD1*, *atrbohF1*, *atrbohD1/F1*, and *atrbohD2/F2*) compared to plants treated only with ABA [170]. Moreover, the double mutant *atrbohD1/F1* has a higher auxin signal compared to WT under ABA exposition (Figure 2C) [170].

BR-ethylene: The *det2-9* mutant accumulates more ethylene and more O_2_^−^ than WT plants in PRs, similar to what happened in WT plants in response to ACC treatment. Moreover, the application of an ethylene biosynthesis inhibitor (AVG) increases the *det2-9* PR growth compared to untreated plants and reduces the O_2_^−^ levels, indicating that ethylene in *det2-9* promotes O_2_^−^ accumulation [166].

In summary, auxin receptors and the signaling pathway of ethylene are important for the inhibitory function of these hormones on PR growth under ROS conditions. Besides, BR synthesis is important for PR growth through H_2_O_2_ and O_2_^−^ homeostasis. Moreover, the inhibition of PR by high ABA levels is partially due to ROS and the PR growth promotion of GA depends on its synthesis and signaling pathway. Finally, the inhibitory effect of ABA increases when the auxin distribution is altered showing the addition of a repressive effect of auxin to the ABA function (Figure 2C and Table 1).

### 3.3. Interaction between Plant Hormones and Metal Toxicity during Primary Root Growth

High concentrations of metals and metalloids in the soil are toxic for plants as they affect its morphology, physiology and biochemistry and, at the cellular level, could induce cell toxicity by the compound itself and, in many cases, by increasing ROS production [172]. Thus, plants have strategies to diminish metal toxicity; in particular, the roots respond to metal contamination by regulating their uptake, retaining them intracellularly in its vacuoles and by precipitating or expelling them as anions. Moreover, phytohormones also participate in this response against injurious agents like heavy metals and metalloids stress [172,173].

#### 3.3.1. Arsenic (As) Toxicity

As and its oxidized states, arsenite (As(III)) and arsenate (As(IV)), negatively affected PR length (Figure 3A) [123,174]. In WT plants, 10 µM As(III) can achieve 50% inhibition of PR growth, whereas As(IV) requires higher concentrations (1.5 mM) to reach a similar degree of inhibition [174].

Auxin: The inhibition of PR growth of mutants that participate in auxin transport (*aux1*, *pin1* and *pin2/eir1-4*) as well as plants treated with auxin transport inhibitors like 1-naphthoxyacetic acid (NOA) and NPA, is stronger in the presence of As(III) than untreated plants [123,174]. Moreover, the exogenous IAA application in *aux1* plants under As(III) treatment increases PR growth, but this is not observed in WT root [123]. Moreover, *pin2/eir1-4* mutant accumulates higher concentrations of As(III) in roots than WT; also, PIN2 and As(III) distributes in highest proportion in the same cells and this distribution is altered in *pin2/eir1-4* mutant. All these data suggest that PIN2 could function as an arsenite efflux transporter. Consequently, the overexpression of *PIN2* increases PR elongation under As(III) stress compared to WT plants [174] (Figure 3A and Appendix A).

#### 3.3.2. Aluminium (Al) Toxicity

Al is the most abundant metal on earth; the concentration of this metal has a biphasic impact in PR growth; low concentration promotes while high concentration inhibits PR growth [175] (Figure 3B).

Auxin: Al promotes the accumulation of auxin in the root and exogenous application of auxin inhibits PR growth in Al stress conditions [175,176]. Co-treatment of Al with NAA inhibits PR growth, whereas the co-treatment with the auxin antagonist α-(phenylethyl-2-one)-indole-3-acetic acid (PEO-IAA) or the auxin transport inhibitor NPA alleviates the PR growth inhibition in response to Al stress. Furthermore, the exogenous application of yucasin, a chemical inhibitor of YUC activity, also alleviates PR growth inhibition by Al [175,177,178]. Accordingly, the PR growth of loss of function mutants in auxin synthesis (*tryptophan aminotransferase of arabidopsis 1* (*taa1-1*), *yuc9* and the double mutant *yuc8 yuc9*); the gain of function mutant in auxin signaling (*solitary root 1* (*slr-1/iaa14*)) or the loss of function mutants in auxin transport (*aux1-7* and *pin2*), have longer PRs under Al stress conditions than WT plant [175,177,178]. Consistently, high endogenous IAA levels obtained with a dominant *YUC1* mutant (*yuc1D*) or with the OE of *PIN8* have stronger inhibition of PR growth than WT plants in response to Al (Figure 3B and Appendix A) [175]. Curiously, the PR growth of ARF mutants (*arf1*, *arf6*, *arf8*, *arf9*, *arf10*, and *arf16*) that has been considered as activators or repressor of auxin signaling and the double mutants, in either ARF activators (*arf7/19*) or repressors (*arf10/16*), are larger than WT plants exposed to Al (Figure 3B and Appendix A) [175]. It would be interesting to demonstrate if these TFs play these activator-repressor roles in PR growth under Al stress condition.

CK: Exogenous CK application inhibits PR growth in Al stress treatment [176]. The PR growth of double and quadruple loss of function mutants in CK synthesis (*ipt3 ipt7*, *ipt5 ipt7*, *ipt1 ipt3 ipt5 ipt7*) and signaling (the single mutants *ahk2*, *cre1* and *ahk2* and the triple mutant *arr1 arr10 arr12*) are longer under Al stress than WT plants [176]. Accordingly, the PR growth of the OE of *ARR1* or *ARR12* and the loss of function mutant of CK catabolism (*ckx3-1*) are shorter than WT plants in response to Al stress [176] (Figure 3B and Appendix A).

Ethylene: Exogenous ethylene treatment inhibits PR growth under Al stress [176,177] and the PR inhibition by Al is alleviated in the presence of inhibitors of ethylene biosynthesis like AVG or Cobalt (Co^2+^) or by the presence of an ethylene antagonist Ag^+^ [177]. Likewise, the PR root length of the loss of function mutants in ethylene signaling (*etr1-3 ein2-1* and *ein3-1 eil1-1*) are longer whereas the *eto1-1* mutant is shorter in Al stress than WT plants [176,177,179] (Figure 3B and Appendix A).

JA: JA intensifies the PR growth inhibition induced by Al since exogenous applications of JA with Al leads to shorter PRs than plants treated only with Al [180]. Accordingly, *coi1-2* and *myc2-2* single loss of function mutants, that participate in JA signaling, and the loss of function mutants of the *ALLENE OXIDE SYNTHASE* (*AOS*) gene, which is part of JA synthesis, have longer PRs than WT plants under Al exogenous application [180] (Figure 3B and Appendix A).

SA: SA is a negative regulator of PR growth in response to Al. The PR root length of the loss of function mutant (*npr1-1*) of the SA receptor *NON EXPRESSOR OF PR GENES 1* (*NPR1*), is less inhibited in the presence of Al exposure compared to WT plants [179] (Figure 3B and Appendix A).

#### Crosstalk under Al Stress Treatment

Auxin-ethylene: Exogenous ethylene inhibits PR growth under Al stress which can be partially reverted when is combined either with an inhibitor of YUC activity (yucasin), with the single mutants *yuc9* and *taa1-1* or with the double mutant *yuc8 yuc9* [175,178]. Accordingly, the inhibitor of TAA1/TAR-dependent auxin biosynthesis, l-kynurenine (Kyn), alleviates the PR growth inhibition generated by the ethylene precursor ACC and Al co-treatment [175]. Moreover, the application of NAA in *eto1-2* or *ein3-1 eil1-1* mutants enhances PR inhibition induced by Al [175]. In addition, auxin accumulation as well as the transcript accumulation of *AUX1* and *PIN2* are increased in treatments with either Al or ACC [177]. All these data indicate that auxin response is downstream of ethylene in Al stress response (Figure 3B).

Auxin-CK: The synthetic cytokinin 6-benzyladenine (6-BA) inhibits PR growth in response to Al stress in single and double auxin biosynthesis or signaling mutants, *taa1-1*, *slr-1/iaa14*, *arf7 arf19*, and *arf10 arf16* [176]. In addition, in the CK biosynthesis or signaling mutants, *ipt5 ipt7* and *arr1 arr10 arr12*, the PR growth inhibition is strongly attenuated, compared to WT plants, under the co-treatment of NAA with Al [176] (Figure 3B).

JA-auxin: JA and auxin function in parallel to regulate the inhibition of PR growth induced by Al. The application of JA in the double loss of function mutant *arf7arf19* or in the single gain of function mutant *slr-1/iaa14* that participates in auxin signaling, leads to shorter PR during Al stress, compared with mutant plants exposed to Al only [180]. Furthermore, the *arf7arf19 coi1-2* triple mutant has longer PR compared with either the JA receptor single mutant, *coi1-2*, the *arf7 arf19* double mutant and the WT plants under Al stress [180].

CK-ethylene: The PR growth of the triple mutant in CK signaling (*arr1 arr10 arr12*) treated with Al is less inhibited than WT plants, but the co-treatment with ACC does not affect PR growth as in WT plants [176]. Co-treatment of Al with 6-BA in the ethylene signaling double mutant (*ein3 eil1-1*) strengthened the Al stress-induced PR growth inhibition suggesting that CK acts downstream of ethylene [176] (Figure 3B).

JA-ethylene: JA acts downstream and in parallel with ethylene to inhibit PR growth in response to Al. In the *ein3-1 eil1-1* double mutant, which partially alleviates root inhibition by Al compared to WT plants, the application of Methyl jasmonate (MeJA) enhances PR inhibition induced by this metal [180]. Moreover, the JA receptor mutant *coi1-2* has larger PR growth than WT plants under Al stress. However, the PR is shorter in the co-treatment with the precursor of ethylene biosynthesis, ACC, and Al in comparison with plants treated exclusively with Al. These results indicated that the inhibition of the PR under Al treatment is mediated by ethylene, acting upstream of JA [180] (Figure 3B).

SA-ethylene: The PR of the single mutants, *ein2-1* and *npr1-1* are less inhibited by Al than WT plants. However, the PR growth of the double mutant *ein2-1 npr1-1* is strongly repressed by this metal suggesting that SA and ethylene have individual as well as common targets that are important for promoting PR growth under this stress condition [179,181] (Figure 3B).

#### 3.3.3. Cadmium (Cd) Toxicity

Cd is also a non-essential toxic heavy metal that inhibits PR growth through a decrease in the size and cell number of the MZ (Figure 3C) [182,183].

Auxin: Cd exposure inhibits PR growth decreasing the auxin accumulation in the root tips. The exogenous application of NAA increases the retention of Cd in the PR, avoiding its translocation to shoots [182,184]. Furthermore, the PIN1/3/7 auxin transport proteins contribute redundantly to the root inhibition mediated by this metal; hence, the PR of the triple mutant *pin1 pin3 pin7;* as well as the auxin receptor mutant (*tir1-1*) and the loss of function mutant *axr3/iaa17*, have longer PRs under Cd treatment than WT plants (Figure 3C and Appendix A) [182].

GA: GA alleviates the PR growth inhibition by Cd since the treatment with 5 µM of GA improves the PR growth and reduces the Cd levels in roots, partially through the IRON-REGULATED TRANSPORTER 1 (IRT1) pathway as shown by the co-treatment of GA with Cd in the *irt1* mutant, which does not alleviate PR growth as happens in WT plants [185] (Figure 3C).

BR: Contrary to GA, reducing BR levels improves Cd tolerance in roots. Cd-induced PR growth inhibition is significantly enhanced in the presence of the synthetic brassinosteroid, epibrassinolide (eBL), whereas PRs treated with the BR biosynthesis inhibitor brassinazole (BRZ), exhibit reduced sensitivity to Cd [186]. Consequently, in response to Cd, the PR of the mutant of BR signaling (*bri1*) is less sensitive to this metal compared to WT [186] (Figure 3C and Appendix A).

Ethylene: Ethylene regulates the PR growth inhibition induced by Cd. The PR growth inhibition by Cd is enhanced when it is co-treated with ACC, whereas the addition of the inhibitor, AVG, diminishes this repression [187]. The ethylene insensitive mutant *ein3-1 eil1-1*, which participates in ethylene signaling; the ethylene receptor mutant *ein4-1;* and the OE of *EIN3-BINDING F BOX PROTEIN 1* (*EBF1*), a repressor of ethylene signaling, have longer PRs than WT plants in response to Cd [187,188]. In contrast, the *ebf1-1* mutant, that accumulates more EIN3 protein; the OE of *EIN3* and *EIL1;* and the mutant of *CONSTITUTIVE RESPONSE 1* (*CTR1*) gene (*ctr1-1*), a negative regulator of the ethylene signaling pathway which leads to an ethylene constitutive response; all of them with elevated ethylene signaling, have a severe PR growth inhibition compared to WT plants in response to Cd [187] (Figure 3C and Appendix A).

JA: Cd exposure promotes the expression of genes that induce JA synthesis and increase the root MeJA concentration as a means to cope with the stress induced by this metal. Consequently, decreasing JA levels magnify the Cd toxicity: the JA synthesis mutant *aos*, with reduced endogenous JA levels, not only has a shorter PR length than WT plants in the presence of Cd, but accumulates more of this metal. The positive function of JA against Cd stress is mediated through the expression of genes that encode antioxidant enzymes and the suppression of genes that promote the Cd uptake and its translocation from roots to shoots [189] (Figure 3C and Appendix A).

SA: Cd stress also increases the levels of SA and the transcript accumulation of its biosynthesis-related genes negatively affecting the plant tolerance to this metal [190]. The transgenic line overexpressing *NahG* gene, that codes a salicylate hydroxylase of *Pseudomonas putida* which oxidizes SA to catechol and consequently diminishing SA levels, has longer than WT plants in response to Cd [190]. Moreover, the *nahG* transgenic plants have less levels of H_2_O_2_ and malondialdehyde (MDA), a compound that increases during excessive oxidative stress and enhances the levels of the antioxidant glutathione compared to WT. These results suggest that low SA levels offer protection against Cd stress through a reduction of cellular oxidative stress [190] (Figure 3C and Appendix A).

Interestingly, Cd increases the levels of JA and SA that act differently regarding PR growth and antagonistically in the defense of plants in response to different biotic stress conditions. Moreover, these two hormones have a complex crosstalk and it would be interesting to study this in response to Cd stress [191,192].

#### 3.3.4. Lanthanum (La^3+^) Toxicity

High concentrations of La^3+^ inhibits PR growth in a dose-dependent manner (10–300 µM) (Figure 3D) [193].

Auxin: La^3+^ is a rare-earth metal that inhibits PR growth suppressing auxin transport [194]. La^3+^ inhibits the abundance of auxin carriers and reduces auxin accumulation in the PR, by repressing PIN-mediated auxin transport. The co-treatment of IAA and La^3+^ generates a PR growth less inhibited compared with the treatment of La^3+^ alone. The supplementation of La^3+^ with NPA, the auxin transport inhibitor, does not further inhibit the PR growth compared with only the La^3+^ treatment. However, the PR growth of the mutants of auxin transport *pin1* and *pin4-3* are less sensitive, while the gain of function *axr3-3/iaa17* is more sensitive, to La^3+^ application than WT plants [194] (Figure 3D and Appendix A).

#### 3.3.5. Chromium (Cr) Toxicity

Cr is a metal greatly toxic to plants and its stable form, dichromate (Cr(VI)), inhibits PR length (Figure 3E) [195,196,197].

Auxin: Cr(VI) inhibits PR growth by enhancing auxin signal and distribution through the expression of *AUX1* [195]. The PR of *aux1-7* is longer under Cr(VI) compared to WT plants. Moreover the exogenous application of NPA significantly recovers the PR growth inhibition mediated by Cr(VI) [195]. Additionally, the PR of the gain-of-function mutant *slr1/iaa14* is larger than WT plants even at elevated concentrations of Cr(VI) (100–200 μM) [198] (Figure 3E and Appendix A).

Ethylene: Cr(VI) promotes ethylene synthesis and signaling that, negatively affects the PR growth. In response to Cr(VI) the PR length of the loss of function mutants of *ETR1* (*etr1-3*) and *ein2-1* are longer compared to WT plants [195]. Accordingly, the *eto1-1* mutant, that has more ethylene, is hypersensitive to Cr(VI) and the PR length is shorter than WT plants (Appendix A). Moreover, the co-treatment of WT plants with ACC and Cr(VI) enhances the inhibition of PR growth in response to the metal and, therefore, the co-treatment with the ethylene inhibitor AVG, significantly recovers the Cr(VI) mediated inhibition of PR growth (Figure 3E) [195].

#### Crosstalk under Cr Stress Treatment

Auxin-Ethylene: Ethylene is involved in Cr(VI)-induced auxin accumulation that mediates the inhibition of PR growth. The PR length of the *aux1-7* mutant co-treated with ACC and Cr(VI) is larger than WT plants (Figure 3E) [195].

#### 3.3.6. Copper (Cu) Toxicity

Cu is a heavy metal that inhibits PR growth at high concentrations (10% in 25 µM CuSO_4_ and up to 68% at 60 µM CuSO_4_) affecting both, the size of the MZ and the size of the EZ (Figure 3F) [81,199,200].

Auxin: Cu excess causes an increase of the auxin signal in PR apices and auxin redistribution, which ultimately inhibits root length by changing *PIN1* expression, therefore *pin1* mutant has longer PRs than WT under Cu stress conditions [199] (Figure 3F and Appendix A).

ABA: There is almost no PR growth in plants growing under Cu excess conditions (25 µM) in co-treatment with ABA. Moreover, the PR growth of loss of function mutants of the high affinity plasma membrane Cu transporters *COPT*, *copt2* or *copt1 copt2 copt6*, are more affected by ABA treatment than WT plants. Accordingly, the OE of *COPT1* is less affected by ABA exogenous application than WT plants [201]. Also, in the co-treatment of Cu with ABA, the mutant of the *SHORT-CHAIN DEHYDROGENASE/REDUCTASE* (*SDR/ABA2*) gene, which participates in ABA synthesis (*aba2*) that has lower ABA levels and longer PRs than WT plants. Moreover, the double recessive loss of function mutant of the negative regulators of ABA signaling that encode protein phosphatases type 2C, *hab1-1 abi1-2*, which exhibits hypersensitivity to this hormone, has shorter PRs in response to Cu [201] (Figure 3F and Appendix A).

#### 3.3.7. Lead (Pb) Toxicity

Pb is a heavy metal that inhibits PR growth (Figure 3G) [202].

Ethylene: The PR of the loss of function mutants *ein2-1* is more sensitive to Pb and has shorter roots than WT plants with higher Pb content [202] (Figure 3G).

To conclude, all metals have a negative impact on PR development except for Al, where high concentrations inhibit whereas low concentration increase PR growth (Figure 3). The hormones analysed behave differently under the distinct metal exposure conditions (Table 1): (A) As inhibits PR growth altering auxin distribution and auxin function as a positive regulator of PR growth in this stress condition. Auxin transport is important for the promoter effect on PR growth under As stress (Figure 3A and Table 1). (B) In Al stress, all hormones that have been used (auxin, CK, ethylene, JA and SA) have an inhibitory effect on PR growth while auxin is the only one that also has a promoter function (Figure 3B and Table 1). The synthesis, signaling and transport of auxin is necessary for PR growth inhibition under Al stress, whereas ARFs function either as promoters or inhibitors. In addition, CK synthesis, signaling and catabolism and ethylene and JA synthesis and signaling are important for PR growth inhibition under Al stress conditions. Curiously, the ethylene inhibition under Al stress of ethylene signaling mutants can be reversed with less auxin and less CK while the co-treatment of Al with these hormones strengthen the Al stress-induced PR growth inhibition. These data suggest that auxin and CK act downstream of ethylene in response to Al stress conditions. In addition, the CK response also acts downstream of auxin signaling and synergistically regulates the PR inhibition. Moreover, the pair of hormones, JA-ethylene, JA-auxin and SA-ethylene, act in parallel pathways regarding PR growth reduction in response to Al stress (Figure 3B). (C) Cd inhibits PR reducing auxin levels and GA reduces Cd toxicity by negatively regulating the Cd uptake into roots via IRT1. In addition, the auxin transport and signaling pathway, the ethylene and BR signaling pathways and the SA catabolism are involved in the negative regulation of PR growth inhibition induced by Cd. Moreover, JA synthesis and GA exogenous application are involved in alleviating the PR growth inhibition by this metal (Figure 3C and Table 1). (D) In La^3+^ stress, auxin has a positive and a negative effect in PR growth that depends on whether the mutants are in auxin transport or in the signaling pathway genes (Figure 3D). (E) Cr(VI) increases auxin accumulation and polar transport reducing PR growth and ethylene promotes auxin accumulation in response to Cr(VI) stress. The signaling pathways of both phytohormones and the auxin transport are involved in PR growth inhibition (Figure 3E and Table 1). In addition, the PR length of the *aux1-7* mutant co-treated with ACC and Cr(VI) is less sensitive than WT plants, suggesting that ethylene is downstream of auxin. (F) Cu in high concentrations repress PR growth due to PIN1-mediated auxin distribution changes. ABA synthesis and signaling pathway participate negatively in the PR growth under Cu excess (Figure 3F and Table 1). (G) Ethylene signaling pathway is necessary for promoting PR growth under Pb stress (Figure 3G and Table 1).

### 3.4. Hormone Function in Primary Root Growth under Different Nutrient Availability Stress

Roots have the ability to explore the soil to obtain the nutrients necessary for plant growth and development and nutrient availability is an important factor that can impact root growth [203].

#### 3.4.1. Nitrogen (N)

N is one of the most essential elements that restrict plant productivity and plants have developed strategies to avoid N limitation and to optimize its use during their life cycle [204]. Nitrate (NO_3_^−^) is the major form of inorganic nitrogen source for Arabidopsis and, in many cases, at 5mM concentration, is able to induce PR growth by increasing meristem cell number and the length of completely elongated cells [205,206]. This effect probably depends on the ecotype used or on the methodological procedures [207,208]. There are other N forms like ammonium (NH_4^+^_) that inhibit PR growth [205,209] (Figure 4A). Also, most nutrient-deficiency responses of roots depend on the status of the nutrient in the whole plant rather than on its external concentration. Interestingly, the nutrient movement in the plant could function to report its sufficiency or scarcity [204].

Auxin: Under low and high N conditions, the IAA treatment reduces WT PR length compared to plants grown under control conditions. However, this reduction is greater under N deficiency [210]. Curiously, the “transceptor” (transporter/sensor) of NO_3_^−^, NPF6.3/NRT1.1 (a member of the NRT1/PTR family), can facilitate the uptake of auxin under low NO_3_^−^ concentrations (Figure 4A) [211]. Also, the PR growth of the auxin receptor mutant, *afb3-1*, is longer in nitrate (5 mM) under hydroponic conditions than WT plants [208]. Moreover, in a medium with ammonium (3 mM (NH_4_)_2_SO_4_), the PR of auxin transport mutants **(***aux1* and *pin2*) grow longer than WT [212] (Figure 4A and Appendix A).

CK: The synthesis of CK in the root and its translocation to the shoot transmits important information about soil conditions; plants that were impaired in CK biosynthesis or in root-to-shoot CK translocation have problems dealing with uneven nitrate availability. Moreover, the root derived CK causes a major transcriptional reprogramming in shoots [213]. In soil with different NO_3_^−^ concentrations, the CK synthesis is promoted in roots and then translocated to the shoot, where it coordinates growth (Figure 4A) [204,214]. Furthermore, in a medium with N-deficient or high N levels, the co-treatment with either trans zeatin (tZ) or cis zeatin (cZ), leads to an important decrement of the PR length compared to control conditions. Interestingly, the tZ treatment causes a major root length reduction than the cZ treatment, in both conditions [210]. Also it has been described that nitrate is able to stimulate PR growth increasing CK biosynthesis and signaling and promoting meristem activity [205]. Accordingly, the PRs of the double mutants in CK signaling (*ahk2 ahk3*, *ahk2 ahk4* and *ahk3 ahk4*) or single mutants in CK synthesis (*ipt3* and *ipt5*) growing under constant nitrate supply are shorter than WT plants [205]. Furthermore, the root growth inhibition of the double mutant *ahk2 ahk4* is due to reduction in cell meristem number and cell elongation in the presence of NO_3_^−^ [205] (Figure 4A and Appendix A).

GA: Under low (50 µM) and high (2500 µM) N conditions, the GA3 treatment increases the WT PR length compared to control conditions without the hormone treatment (Figure 4A). However, the PR length is greater under the co-treatment of GA3 and high N [210]. Moreover, the OE of the nitrate transporter of the NRT1/PTR FAMILY (*NPF3*), a low-affinity nitrate/nitrite transporter that can also transport GA in the root endodermal cells in the EZ, has shorter roots than WT plants. Interestingly, this growth inhibition is due to the retention of GA at the sites of synthesis avoiding its movement to the growth sites; accordingly, the GA exogenous application stimulates root growth of *NPF3* OE line [215]. In addition, *NPF3* is a gene that is repressed by GA and functions as an active importer as shown by in vivo and in vitro experiments [215].

BR: There is a polymorphism (proline to leucine substitution) in the kinase domain of the BR signaling, BRASSINOSTEROID-SIGNALING KINASE 3 (BSK3), of 56 Arabidopsis accessions that improves BR sensitivity and signaling and that increases PR growth under mild N deficiency [216]. Moreover, under low N levels, the loss of function mutant of *BSK3* gene (*bsk3*) has shorter PRs than WT plants. Likewise, the double (*bsk3 bks4*), triple (*bsk3 bsk4 bsk7* and *bsk3 bsk4 bsk8*), and the quadruple mutants (*bsk3 bsk4 bsk7 bsk8*) have shorter roots compared to WT plants, either under low (0.55 mM) or high (11.4 mM) N (Figure 4A and Appendix A). Interestingly, the length of the PR is reduced by changes in the length of the completely elongated cells without altering the meristem size under both N growth conditions [216].

ABA: NPF3 can also transport ABA, an antagonist of GA in different developmental processes and, contrary to the phenotype described above for GA, the ABA exogenous application inhibits PR growth in the OE line of the nitrate transporter *NPF3*. Moreover, ABA induces its expression [215] (Figure 4A).

Ethylene: The PR growth of plants growing in the co-treatment with ACC in a medium with low or high N levels are shorter than WT plants [210] (Figure 4A).

JA: In a medium with low or high N levels, the co-treatment with MeJA leads to a decrement in the PR length compared to control conditions [210] (Figure 4A).

SA: Under low and high N starvation, the SA co-treatment reduces WT PR length compared to plants grown under control conditions. In addition, in the *salicylic acid induction deficient 2* (*sid2-2*) mutant, which encodes an isochorismate synthase necessary for SA synthesis, or the transgenic OE line *nahG*, which degrades SA, the PR growth is less reduced under the co-treatment of SA and high or low N concentration than in WT plants compared to control conditions [210] (Figure 4A and Appendix A).

#### 3.4.2. Phosphate (P)

Inorganic phosphate (Pi) is another essential nutrient that is required for plant growth and root development. Root response to Pi availability depends on the concentration used; under low concentrations of NaH_2_PO_4_ (0.001 to 0.01 mM) the PR is short and at higher concentrations (0.1 mM to 1 mM) the PR is longer compared to WT plants (Figure 4B) [217,218]. Interestingly, the physical contact of the PR tip with a low Pi (5 µM) medium is enough to inhibit root growth [219]. Moreover, under Pi deprivation, Arabidopsis accessions behave differently. Of 73 accessions studied, half of them reduced root growth under low Pi, a quarter does not respond to phosphate availability, while the rest of the accessions decrease their PR length (16%) [220]. These data indicate the complexity of the molecular mechanisms underlying PR growth response to different Pi concentrations and that the plasticity of the different accessions could be used to explore this.

Auxin: High concentrations of auxin are required to inhibit PR growth in seedlings grown under high Pi conditions (1mM). Moreover, either in low or high auxin concentration, the PR growth is inhibited under low Pi concentrations (Figure 4B) [217]. Accordingly, the OE of *YUC1* that participates in auxin biosynthesis, causes shorter PRs in response to low Pi [221]. In addition, the PR growth is inhibited using an auxin efflux transport inhibitor (2,3,5-triiodobenzoic acid; TIBA) in high and low Pi concentration, but the inhibition effect is more drastic under low Pi concentration [217]. Furthermore, the PR growth response is opposite in auxin transporters mutants (*aux1-7* and *pin2),* since they have longer roots under high Pi conditions than WT [217] (Figure 4B). Also, under high Pi conditions, the OE of *TIR1* and the triple auxin receptor mutant *tir1 abf2 abf3* have shorter PR than WT plants suggesting that the level of expression of auxin receptor modulates nonlinearly PR growth [222]. In addition, the PR growth of two different auxin-resistant mutants behave differently: the gain-of-function mutant of the auxin repressor *axr2-1* (*iaa7*) is longer under high Pi (1mM) [217,223], whereas the gain-of-function mutant of *IAA28* (*iaa28-1*) has shorter roots either in low or high Pi treatment compared to WT [217] (Figure 4B and Appendix A).

CK: Pi starvation reduces the expression of CK receptors [224] and exogenous applied CK inhibits PR growth under low and high Pi concentrations (Figure 4B) [217]. Recently, it has been described that in roots and in response to Pi-starvation, the levels of tZ are reduced and the levels of cZ are increased. Interestingly, PRs treated with tZ are shorter than those treated with cZ, regardless of the amount of Pi in the medium and both plants have shorter PRs than untreated plants (Figure 4B). In addition, the cell length in the TD with Pi-starvation and cZ co-treatment, but not tZ, can reverse the cell length reduction caused only by Pi-starvation [225].

GA: The changes in root architecture under Pi deprivation response are dependent on the DELLAs proteins inhibitory activity. Pi starvation decreases the bioactive GA levels and subsequently increases DELLAs protein accumulation (Figure 4B) [226]. Under low Pi, the quadruple DELLAs signaling mutant (*gai-t6 rga-t2 rgl1-1 rgl2-1*) and the GA-deficient *ga1-3* with the DELLAs mutants (*ga1-3 gai-t6 rga-t2 rgl1-1 rgl2-1*) have longer PRs, whereas in *ga1-t*, a GA-deficient mutant, the PR length is shorter compared to WT in low Pi and can be restored with GA co-treatment. The GA insensitive mutants as *ga insensitive (gai),* that is a gain of function mutant of one DELLA, and *sleepy1-10* (*sly1-10*), that enhance DELLAs accumulation, have shorter PR compared to WT plants during Pi starvation (Figure 4B and Appendix A) [226].

BR: Low Pi levels do not inhibit PR growth in the gain of function mutants of the BR signaling positive regulators, BZR1 and BRI1-EMS-SUPPRESSOR 1 (BES1) (*bzr1-D* and *pUBQ10-bes1-D*), compared to WT roots [227,228]. Moreover, the BR biosynthesis inhibitor (BRZ) enhance PR growth inhibition in WT plants, reaching growth full inhibition at 60 µM Pi; accordingly, BRZ treatment did not affect root sensitivity of *bzr1-D* to low Pi, showing a constant PR growth [228] (Figure 4B and Appendix A).

Ethylene: It has been reported that ACC (0.1 µM to 10 µM) co-treatment with high or low Pi concentrations inhibits PR growth and meristem size compared to control conditions without hormone treatment [217,218]. Interestingly, different PR growth responses occur when plants are co-treated with the ethylene signaling inhibitor 1-methylcyclopropene (MCP) or the ethylene synthesis inhibitor AVG at different Pi concentrations. MCP and AVG increase PR growth under high Pi but decrease PR growth under low Pi concentrations [229]. The effect of AVG in the PR growth could be restored by the co-treatment with ACC (0.4 µM) to nearly the level obtained under high Pi treatment alone and to exactly at the levels obtained with low Pi treatment [229]. Contrary, the use of Ag+, an ethylene perception inhibitor, diminishes the inhibition of PR growth under Pi-deficiency [230,231]. These data suggest that, at low Pi concentrations, the different ethylene inhibitors used, could have another function that seems to be partially independent of ethylene over PR growth. In addition, under low or high Pi, the OE of the high-affinity Pi transporter *Pht1;5* significantly diminishes the PR length compared to WT and this phenotype is reversed by application of AVG [232,233]. Moreover, *eto1* and *ctr1*, which lead to an ethylene constitutive response and the loss of function mutant of the hypersensitive to Pi starvation 3 *HSP3/ETO1* gene (*hsp3-1* and *hsp3-2*), which overproduces ethylene, have shorter PRs either in low or high Pi concentrations compared to WT [217,234]. As expected, the application of AVG or Ag+, abolishes the inhibition of PR growth of *hsp3-1* [235] (Figure 4B and Appendix A).

JA: Under low Pi, the *low phosphorus insensitive* (*lpi4*) mutant is less inhibited by exogenous JA application than WT plants [218]. Moreover, the PR growth of the loss of function of 12-*OXOPHYTODIENOATE REDUCTASE 3* (*OPR3*), a gene responsible for JA biosynthesis, is longer under Pi deficiency than WT plants independently of *COI1*. In addition, exogenous application of JA reduces PR growth in *opr3*, but the PR length is still longer than WT in response to low Pi [236] (Figure 4B and Appendix A).

SL: At low Pi-conditions, the SL signaling *max2-1* mutant has less PR reduction over time than WT plants [237] (Figure 4B and Appendix A).

##### Phytohormone Crosstalk under Pi Stress Conditions

GA-JA: The negative regulation of PR growth by *OPR3* under Pi deficiency is dependent on GA accumulation, because the JA synthesis mutant (*opr3*) has more GA synthesis and less degradation of this hormone, leading to long PR under this stress. Therefore, under low Pi supply, the application of exogenous GA stimulates PR growth in WT plants, but not in *opr3* mutants [236]. (Figure 4B).

JA-ethylene: *OPR3* interacts with ethylene pathway to control PR growth in response to P limitation, since co-treatment with AVG (ethylene synthesis inhibitor) or Ag+ (an ethylene perception inhibitor) promote PR growth in WT plants in response to Pi deficiency but not in the PR of the *opr3* mutant [236] (Figure 4B).

#### 3.4.3. Potassium (K)

K is an essential macronutrient and the main cation in plant cells. Moreover, it is essential for various enzymatic activities and for plant growth; and is also important to withstand osmotic stress conditions. The PR length decreases as the K concentration in the medium cultures is diminished (1025, 1600, 250, 0 µM) (Figure 4C) [203,238,239].

Auxin: Low K stress diminishes auxin concentrations (Figure 4C), reduces PIN1 protein levels and triggers its degradation and the NAA treatment alleviates low K inhibition in WT plants (Figure 4C) [240]. However, the PR growth of the mutant in the K+ channel *AKT1* (*Arabidopsis K+ transporter 1*), that is involved in the perception of K concentrations and that can still grow under low K compared to WT, is inhibited under exogenous auxin treatment [240]. Likewise, the loss of function of the K carrier, *TRH1* (*trh1*) has shorter PR in response to exogenous auxins compared to WT in control conditions [241]. Besides, both transporters are involved in auxin transport regulation [240,241].

CK: Under K-starved conditions the PR length is reduced, which correlates with a decrease in CK levels compared to K sufficient conditions [238]. Interestingly, the OE of *IPT3*, reduces PR growth under K-deficient and the loss of function *ipt1*,*3*,*5*,*7* is insensitive to K deficiency for 7 days, compared to K sufficient conditions (Figure 4C) [238]. Accordingly, the PR growth of the CK receptor mutants are less affected by K starvation than WT plants but in different degrees: *ahk2* and *ahk3* single and *ahk2 ahk3* and *ahk3 ahk4* double mutants are not affected by K-starved conditions, whereas *akh4* single and *ahk2 ahk4* double mutants, are less affected than WT, but more affected than the other CK receptor mutants under K-deficiency compared to K-sufficient conditions [238] (Appendix A).

Ethylene: K deprivation in roots induces ethylene production through the upregulation of genes implicated in ethylene biosynthesis and signaling [242]. Under low K stress, PR growth is repressed in WT plants but not in *etr1-1* and *ein2-1* [243] (Figure 4C and Appendix A).

#### 3.4.4. Boron (B)

B is an essential micronutrient for plants that limits crop productivity worldwide. Under B deficiency treatments (1, 0.4, 0 µM) the length of the PR decreases significantly (Figure 4D), specifically affecting cell elongation [203,244,245].

Auxin: Under sufficient B levels, auxin is produced in the root tip and transported to the EZ where it remains without an inhibitory effect. However, at low B conditions, the auxin distribution in the PR tip is affected due to the different function of auxin transporters, *AUX1* and *PIN2*; in auxin influx carrier mutant, *aux1-22*, the PR is longer, whereas in the auxin efflux carrier mutant, *pin2* is shorter than WT plants under B-deficiency [244,246] (Figure 4D and Appendix A).

CK: In response to B deficiency the CK perception and CK signaling are altered: the double *ahk2 ahk3*, *arr1 arr12* and triple *arr1 arr10 arr12* mutants showed PR shorter than WT plants [247] (Figure 4D and Appendix A).

BR: B deficiency downregulates BR signaling that ultimately inhibits PR growth. In B-deficient conditions the PR growth is significantly enhanced by the exogenous application of the synthetic BR (eBL), while application of BR biosynthesis inhibitor (BRZ) aggravated PR growth inhibition of B-deficient conditions. Moreover, in mutants of BR signaling, like the gain-of-function mutant *bes1-D*, which over-accumulates the dephosphorylated active BR-regulated TF *BES1*, has longer PR length compared to WT under B deficiency condition. In contrast, under B sufficiency, the BR receptor mutant *bri1-301* has a shorter PR than WT plants [248] (Figure 4D and Appendix A).

ABA: When B is low, B is carried into roots by the transporter NODULIN26-LIKE INTRINSIC PROTEINS 5;1 (NIP5;1). ABA treatment decreases the endogenous B levels in the mutant *nip5;1-1* when the plant is transferred from a B deficient medium to controlled conditions [249]. Moreover, unlike PR growth inhibition in WT plants, the loss of function mutant *nip5;1-1* does not behave differently when the ABA inhibitor Nordihydroguaiaretic acid (NDGA) is applied in plants grown under control conditions and transferred to B deficiency. Furthermore, the PR growth of WT plants grown in B deficiency is inhibited either by ABA or NDGA, while the PR growth of *nip5;1-1* is unaffected [249] (Figure 4D).

#### 3.4.5. Iron (Fe)

Fe is an essential nutrient that is required during the photosynthesis in the electron-transport chain and it is part of the prosthetic group of many enzymes [250]. Low Fe condition has a promoting effect in WT PR growth [203,227], while an excess of Fe limits PR growth by declining cell elongation and division [251] (Figure 4E).

Auxin: Excess of Fe (350 µM) inhibits PR growth and causes an increase of auxin signal in PR apex compared with control conditions (50 µM) (Figure 4E). Furthermore, under Fe stress, the application of IAA significantly reduces the PR growth compared to control plants (Figure 4E) [252]. In response to Fe excess, the auxin transport mutants, *aux1-7* and *pin2-1*, show a decrease in PR growth compared to WT plants probably due to the perturbation of auxin distribution (Figure 4E) [252].

CK: Co-treatment of 6-BA with sufficient or deficient Fe levels inhibit PR growth more than plants treated only with Fe (Figure 4E) [253]. Additionally, CK downregulates the expression and protein accumulation of the Fe transporter *IRT1* [253] (Figure 4E).

GA: Fe inhibits PR growth, in part, via a DELLAs-dependent pathway. The PR growth of the loss of function of the quintuple DELLAs mutant (*gai-t6 rga-t2 rgl1-1 rgl2-1 rgl3-4*) is less inhibited in Fe-deficient conditions than WT plants [254] (Figure 4E and Appendix A).

BR: Low Fe suppresses PR growth inhibition of roots treated with the BR-biosynthesis inhibitor BRZ, suggesting that the inhibitory activity of low BR is relieved by low Fe [227]. The PR growth in the mutants of the cell-wall-targeted ferroxidases *LOW PHOSPHATE ROOT1* (*lpr1*) and the BR signaling gain of function mutants, *brassinosteroid kinase inhibitor 1-1* (*bki1-1*) and *bzr1-D,* are longer than WT plants treated with low Fe, while PR growth of the mutant *bri1* is shorter than WT plants in response to low Fe. Also, *lpr1* is less sensitive to the BR inhibitor BRZ, while the constitutive expression of *LPR1* by *pUBQ10-LPR1* is more sensible to BRZ [227] (Figure 4E and Appendix A).

Ethylene: High levels of Fe promotes ethylene accumulation, which induce PR growth tolerance, since ACC co-treatment reverses PR growth inhibition caused by excess Fe, whereas ethylene inhibition reduces PR length, root cell elongation and cell division seen under Fe excess [251]. In addition, the ethylene-overproduction mutants, *eto1-1*, *eto2-1* and *ctr1-1*, have increased PR growth than WT plants in response to Fe; accordingly, *etr1-3* reduces PR growth in response to Fe [251] (Figure 4E and Appendix A).

In summary, different nutrient availability has diverse effects on PR growth and, as with metal exposure, the different hormones behave differently (Figure 3 and Figure 4 and Table 1): (A) While NO_3_^−^ induces PR growth, NH_4^+^_ represses it. Auxin transport and signaling and SA synthesis or catabolism participate in PR growth inhibition, while CK synthesis and signaling are required to induce PR growth under constant and high levels of NO_3_^−^. In addition, BR signaling is required to induce PR growth under either low or high NO_3_^−^ conditions. Besides, exogenous application of CK, ethylene and JA inhibit whereas GA exogenous application induces PR growth under low and high N conditions. Finally, ABA exogenous application inhibits PR growth of the OE of *NPF3* (Figure 4A and Table 1). (B) Low Pi levels reduce PR length while high Pi levels lead to longer PR. High exogenous auxin concentration with high or low Pi inhibits PR growth, whereas low exogenous auxin concentration only with low Pi inhibits PR growth; also auxin synthesis inhibits PR growth under low Pi concentration (Figure 4B). Auxin signaling promotes and inhibits PR growth under high Pi treatment, while promotes PR growth only under low Pi treatment (Figure 4B). In addition, auxin transport also promotes and inhibits PR growth under high and low Pi concentration (Figure 4B). CK exogenous application inhibits PR growth under both low and high Pi concentration whereas GA and BR signaling are necessary for PR promotion under low Pi (Figure 4B). Finally, treatment with high or low concentration of ethylene inhibits the PR growth independently of Pi concentration, although the use of ethylene inhibitors show that ethylene could have a positive role in PR growth in response to high Pi. Moreover, the overproduction of endogenous ethylene or an ethylene constitutive response inhibits PR growth under high and low Pi (Figure 4B). In addition, JA, GA and ethylene participate in a regulatory network, where GA has a positive role and ethylene and JA act as negative regulators of PR growth under low Pi. (C) Low K inhibits PR growth and exogenous auxin application induces PR growth under these growth conditions, while CK synthesis and signaling, and ethylene signaling are required to inhibit PR growth (Figure 4C and Table 1). (D) Low B reduces PR length. Auxin has a dual role on PR growth as PIN2 enhances it whereas AUX1 inhibits it during B deficiency. CK signaling and BR signaling and biosynthesis, are required to stimulate PR growth while ABA inhibits it under B deficiency (Figure 4D and Table 1). (E) The low concentration of Fe increases PR growth, whereas high Fe levels reduce it. Auxin exogenous application reduces PR growth with high Fe co-treatment; interestingly, auxin transport is necessary, under this condition, to stimulate PR growth (Figure 4E). Besides, CK exogenous application also inhibits PR growth under sufficient or deficient Fe-co-treatment and also regulates iron uptake. GA signaling, BR signaling, and ethylene synthesis and signaling induce PR growth under low and high Fe respectively (Figure 4E and Table 1).

### 3.5. Increased Environmental Temperature and Hormone Function during Primary Root Growth

High temperature (26 °C) stimulates PR growth that depends on root cell elongation and not on increased cell proliferation, as the meristem (size and cell number) is shorter than in WT plants [255] (Figure 5A).

Auxin: In roots, high temperature stimulates the auxin production, disturbing the intracellular auxin distribution. In addition, high temperature promotes the downregulation of *PIN-LIKES* (*PILS*) that are negative regulators of auxin signaling, and, consequently, increment auxin nuclear accumulation and PR growth (Figure 5A) [256,257]. At 29 °C, the double mutant in auxin receptors *tir1-1 afb2-3*, the single mutants in auxin transporters *aux1-7*, *pin2/eir1-1* and the OE in the auxin transport *PIL6*, have shorter PR compared to WT plants (Figure 5A and Appendix A) [257,258,259]. Also, at 27 °C, the PR length of the *TAA1* mutant, also called *CK-INDUCED ROOT CURLING1* (*ckrc1-1*), is significantly short compared to WT plants. Exogenous NAA, but not IAA, restores the root growth phenotype of *ckrc1-1* [260].

BR: The level of BRI1 protein decreases under high temperature whereas the level of the transcript does not change [255] and *bri1* BR receptor mutant has longer PRs than WT plants in response to high temperature (26 °C–29 °C). Accordingly, the constitutive BR response mutants, *bzr1-D* and *bes1-D*, and the triple GSK (Glycogen Synthase Kinase) mutant *bin2 sk2-2* (*bil1*) *sk2-3* (*bil2*) (defective in the negative regulation of the BR pathway) have shorter PRs at 26 °C compared to WT plants (Figure 5A and Appendix A) [255,261].

ABA: ABA seems to regulate the plant capacity to deal with lethal high temperatures (45 °C) only when previous acclimatization at a sublethal high temperature (38 °C) is applied (acquired thermotolerance). Moreover, the ABA signaling and gain of function mutants (*abi1*, *abi2*), which are insensitive to ABA and the ABA biosynthesis mutants (*aba1*, *aba2*, and *aba3; ABA3* gene codes a *MOLYBDENUM COFACTOR SULFURASE*) have a strong PR growth reduction compared to WT under high temperature [262] (Figure 5A and Appendix A).

Ethylene: As with ABA, the ethylene signaling mutants (*ein2* and *etr1*) show shorter PR than WT during the acquired thermotolerance conditions [262] (Figure 5A and Appendix A).

SA: Also, during the acquired thermotolerance treatment, the transgenic line *nahG*, which degrades SA, has a reduced PR compared to WT [262] (Figure 5A and Appendix A).

#### Phytohormone Crosstalk under High Temperature

Auxin-BR: BR controls the PILS-dependent nuclear accumulation of auxin. The OE of *PILS5* in *bri1-301* mutant produces a slightly longer PR growth at high temperature than WT plants suggesting the parallel participation of both hormones in PR growth [256]. (Figure 5A).

Auxin-Ethylene: *CKRC1 (TAA1)* is also implicated in the regulation of ethylene signaling [263]. The application of ACC promotes slightly the PR length of *ckrc1-1* at 22 °C and 27 °C. Conversely, at 27 °C, AVG increased severely the root defects of *ckrc1-1* and inhibited PR growth compared to WT plants [260] (Figure 5A and Appendix A).

Summarizing, high temperature stimulates PR elongation; and auxin synthesis and transport positively regulate PR growth in this stress condition. In addition, ABA synthesis and signaling, ethylene signaling and SA catabolism act positively in PR growth during acquired thermotolerance, while BR signaling is necessary for PR growth inhibition during high temperature (Figure 5A and Table 1). Furthermore, ethylene is below auxin in this stress response and auxin and BR act in parallel. Moreover, PILS are negatively controlled by BR and the OE of *PIL5* slightly revert the short PR phenotype of *bri1-1*.

### 3.6. Hormone Participation in Primary Root Growth under Soil Alkalinity

Soil alkalinity restricts agricultural productivity producing plants with shorter PR. The growth rate of PR is significantly lower after 24 h of alkaline stress (pH 8) than at control conditions (pH 5.8) [264]. Moreover, alkaline conditions inhibit Arabidopsis PR growth by decreasing cell division in the MZ via ethylene signaling [265] (Figure 5B).

Auxin: In WT plants, alkaline stress reduces PR length and enhances auxin distribution in the epidermal cells of PR tips due to an increase of the transcript of the auxin transport PIN2 in comparison to control conditions. PIN2 also modulates proton secretion in the root tip to preserve PR elongation under alkalinity [264]. Thus, in response to alkaline stress, the mutant, *pin2* reduces PR growth (Figure 5B) [264]. Interestingly, the PR growth of auxin influx transport and biosynthesis mutants: *aux1-7*, *wei2/asa1* (*ANTHRANILATE SYNTHASE alpha SUBUNIT 1*) and *wei8* (*TAA1*) are less affected by alkaline stress (pH 8 and 9) than WT plants (Figure 5B and Appendix A) [265].

Ethylene: The ACC exogenous application produces a severe diminution of PR growth after alkaline stress treatment compared to control conditions; this phenotype could be partially reversed by the addition of the antagonist of ethylene biosynthesis (Co^2+^) [265]. The PR growth of the ethylene overproducer mutant, *eto1*, is hypersensitive to alkaline stress and has shorter roots compared to WT plants; accordingly, the single mutants of ethylene signaling, *etr1-3*, *ein2* and *ein3-1* have longer PRs under alkaline conditions [265] (Figure 5B and Appendix A).

#### Phytohormone Crosstalk under Soil Alkalinity

Auxin-ethylene: Ethylene regulates the inhibition of PR induced by alkaline stress increasing auxin accumulation (Figure 5B) [265]. The alkalinity-induced *AUX1* expression is inhibited in roots with the treatment of Ag^+^ or Co^2+.^ In addition, *eto1-1* does not affect the sensitivity of *aux1-7* to alkalinity, since the double mutant *eto1-1 aux1-7* has longer PR than WT [265].

In summary, as with B, *aux1-7* has longer whereas *pin2* has shorter PR under alkaline stress. Ethylene synthesis and signaling are required to the negative regulator function of this hormone on PR growth under alkaline stress and the PR growth of the double mutant *eto1-1 aux1-7* demonstrated that ethylene is above auxin in this stress response (Figure 5B, Table 1 and Appendix A).

## 4. Conclusions

The root development phenotypic responses to different stress conditions are highly plastic, and phytohormones provide signals that allow the plants to dynamically respond to them. This review showed the importance of the phytohormones (auxins, CK, ABA, JA, ethylene, GA, SL, and SA) and the crosstalk among them under different stress conditions to determine Arabidopsis primary root growth and plasticity. Interestingly, and despite the hormone function under controlled conditions, it is impossible to predict the hormonal participation in stress-induced primary root changes as shown in the examples presented in this review. The high sensitivity of the root to slight changes of hormone concentration, distribution, and the crosstalk among them or with other internal and external signals, as well as the lack of comprehensive studies, make the interpretation of the hormonal function in stress responses difficult. Surprisingly, all hormones studied, except for GA that always acts as a positive regulator of primary root growth in response to stress conditions described here, could either promote or inhibit primary root growth depending on the abiotic stress conditions (Table 1 and Figure 2, Figure 3, Figure 4, Figure 5). Moreover, their final function change depends on the spatiotemporal dynamics of the hormones, whether they interact synergistically or antagonistically with other hormones and on the different developmental stages studied. As shown in this review, Arabidopsis mutant analysis of either loss or gain of function mutant of the different genes that participate in the hormone homeostasis and/or the exogenous application of compounds that change the hormone concentration has helped in the integration of the phytohormone crosstalk function in primary root growth, but it would be important to deepen in the participation of various inputs added simultaneously, to integrate the phenotypic response.

## Figures and Tables

**Figure 1 cells-09-02576-f001:**
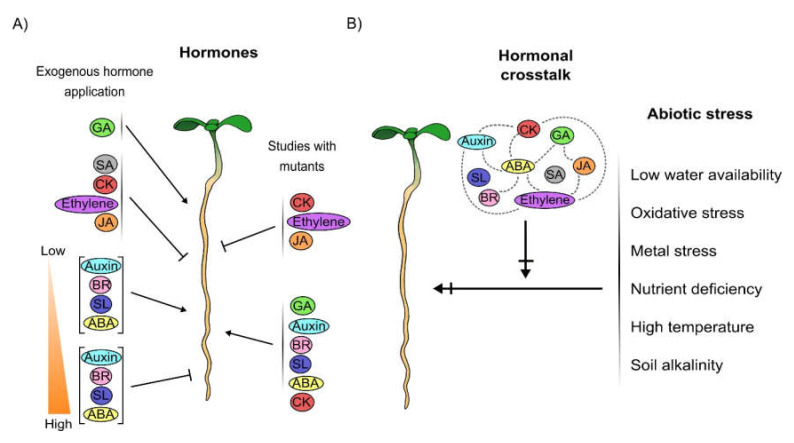
Hormone function under controlled conditions affecting primary root growth and its participation in abiotic stress response. Each hormone is represented by a different color; black arrows represent promotion whereas blunt arrows represent inhibition of primary root growth. (**A**) Hormone participation in primary growth using either external exogenous application or loss or gain of function mutants. (**B**) Interplay between abiotic stress as the external stimulus, and hormone participation as the internal stimulus, on primary root growth and dotted lines represent the interaction of hormones under different abiotic stress conditions and their crosstalk affecting primary root growth.

**Figure 2 cells-09-02576-f002:**
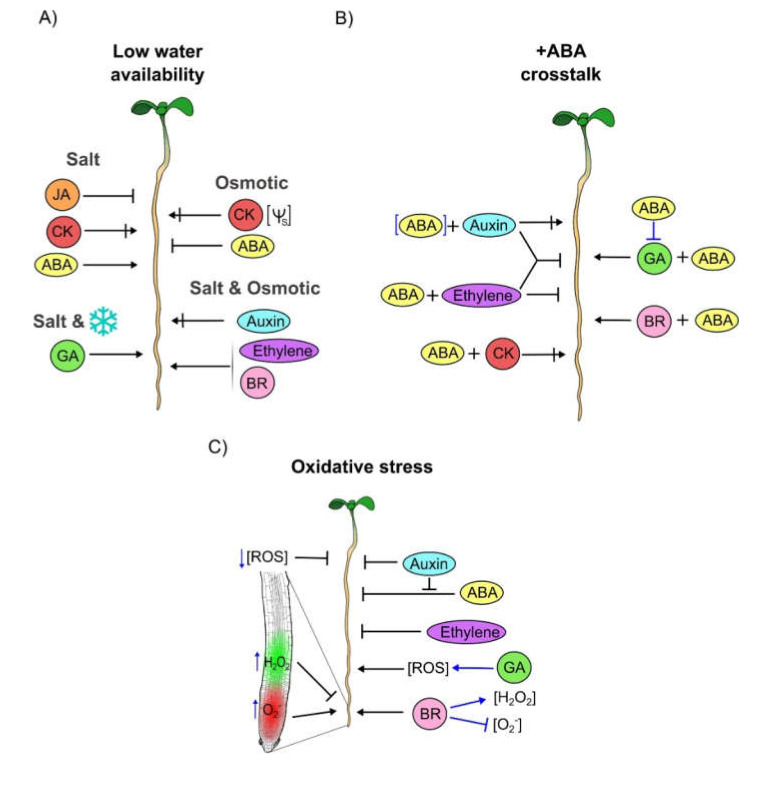
Hormone function on primary root growth under low water availability, the crosstalk with ABA, and oxidative stress conditions. Each hormone is represented by a different color; black arrows represent promotion whereas blunt arrows indicate inhibition of primary root growth. Blue arrows represent regulation of hormone levels, ROS concentration or inhibition or promotion of different ROS. (**A**) The effect on primary root growth of each hormone under osmotic, osmotic and salt, salt and cold, and salt stress is shown. [Ψs] is the osmotic potential of the medium that affects CK function on primary root growth and 
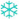
 is the symbol for cold stress. (**B**) ABA interaction with other hormones affecting primary root growth, was symbolized by a “+”. (**C**) H_2_O_2_ inhibits while O_2_ promotes primary root growth, besides, O_2_ is distributed in the meristematic zone and H_2_O_2_ in the elongation zone.

**Figure 3 cells-09-02576-f003:**
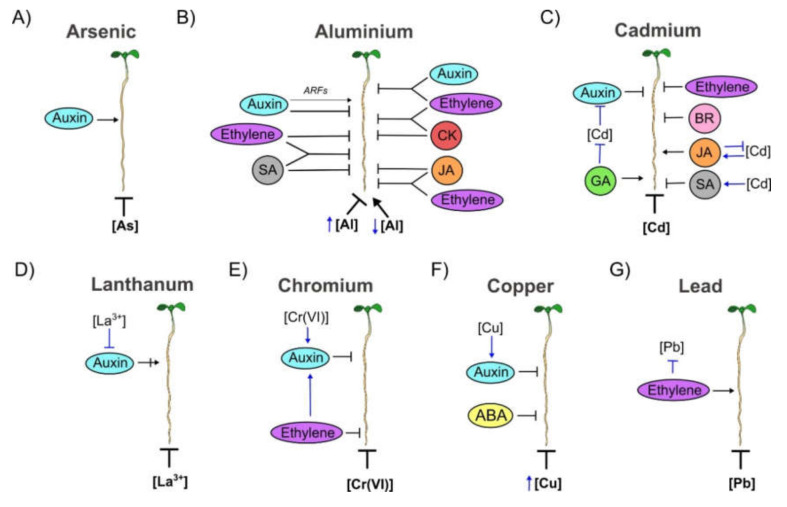
Hormone function on primary root growth under different metal stress conditions. Each hormone is represented by a different color; black arrows represent promotion whereas black blunt arrows show inhibition of primary root growth. In addition, blue arrows represent regulation of hormone levels and metal concentration. At the bottom of each figure is the overall effect of different concentrations of each metal on primary root growth: arsenic (**A**), aluminium (**B**), cadmium (**C**), lanthanum (**D**), chromium, (**E**) copper (**F**) and lead (**G**). In aluminium stress conditions, different ARFs, that have been described as activators or repressors of auxin signaling, have a negative effect over primary root growth and are marked over the black arrow.

**Figure 4 cells-09-02576-f004:**
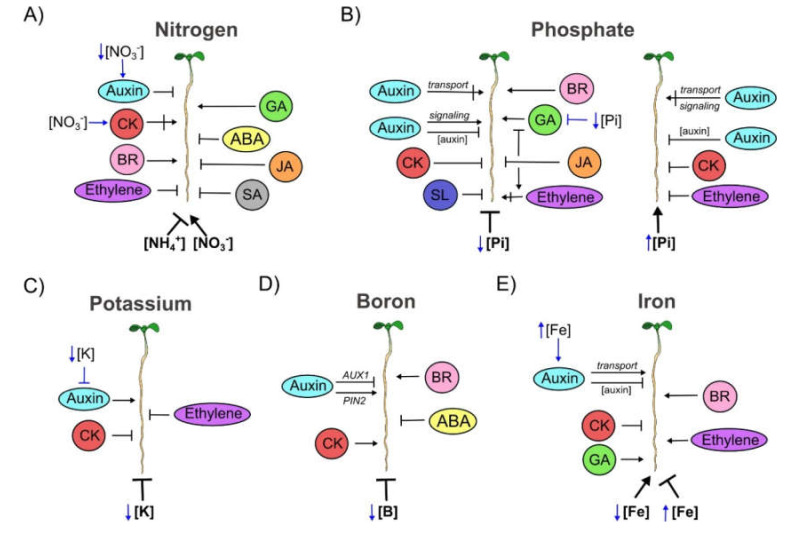
Interaction between plant hormones and different nutritional conditions over primary root. Each hormone is represented by a different color; black arrows represent promotion whereas black blunt arrows show inhibition of primary root growth. In addition, blue arrows represent regulation of hormone levels and nutrient concentration. At the bottom of each figure is the overall effect of different concentrations of each nutrient on primary root growth: nitrogen (**A**), low and high phosphate (**B**), potassium (**C**), boron (**D**) and iron (**E**). Primary root growth promotion or inhibition in auxin transport, signaling and concentration is marked. In (**B**) the phosphate concentration is divided into low and high to emphasize hormone function in PR growth under these stress conditions.

**Figure 5 cells-09-02576-f005:**
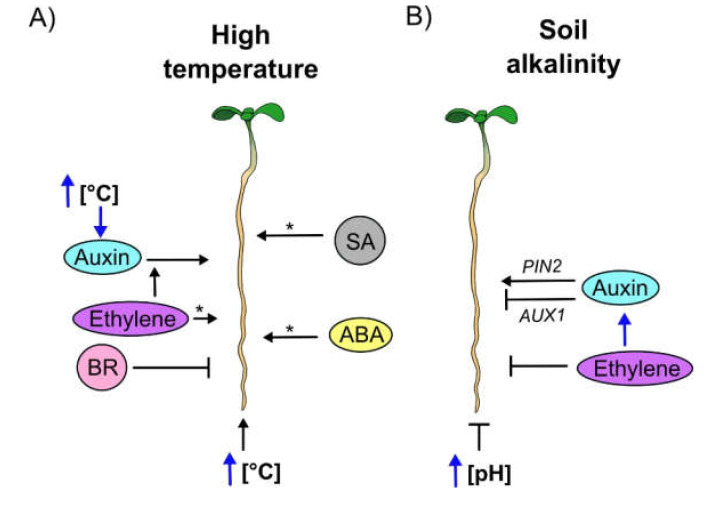
Hormones implicated in primary root growth under high temperature and soil alkalinity. Each hormone is represented by a different color; black arrows represent promotion whereas black blunt arrows depict inhibition of primary root growth. In addition, blue arrows represent regulation of hormone, temperature and pH levels. Asterisk refers to thermotolerance response. (**A**) High temperature stimulates primary root growth. (**B**) Alkalinity inhibits primary root growth. Primary root growth promotion or inhibition in auxin transport is marked.

**Table 1 cells-09-02576-t001:** Function of hormones in the primary root development under different stresses.

	Auxin	CK	GA	BR	ABA	Ethylene	JA	SA	SL
Osmotic	**+/−**	**+/−**		**+**	**-**	**+**			
Saline	**+/−**	**+/−**	**+**	**+**	**+**	**+**	**−**		
Cold			**+**						
ABA	**+/−**	**+/−**	**+**	**+**		**−**			
Oxidative	**−**		+	+	**−**	**−**			
Arsenic	**+**								
Aluminium	**+/−**	**−**				**−**	**−**	**−**	
Cadmium	**−**		**+**	**−**		**−**	**+**	−	
Lanthanum	**+/−**								
Chromium	**−**					**−**			
Copper	**−**				**−**				
Lead						**+**			
Nitrogen	**−**	**+/−**	**+**	**+**	**−**	**−**	**−**	**−**	
Phosphate	**+/−**	**−**	**+**	**+**		**+/−**	**−**		**−**
Potassium	**+**	**−**				**−**			
Boron	**+/−**	**+**		**+**	**−**				
Iron	**+/−**	**−**	**+**	**+**		**+**			
High temperature	**+**			**−**	**+**	**+**		**+**	
Alkalinity	**+/−**					**−**

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
