# Peer review of "Interplay between Hormones and Several Abiotic Stress Conditions on Arabidopsis thaliana Primary Root Development"

_cells, 2020, doi:10.3390/cells9122576_

Round 1

Reviewer 1 Report

Dear authors,

your review summarized lot of investigations on Arabidopsis root development in respornse to environmental conditios, hormones and hormonal crosstalk in stress conditions. The data collected in your review will be very helpfull for the researchers working in the field of Arabidopsis root development.

I have few remarks that need to be improved.

Abstarct page 3 line 87 change to has high phenotipic plasticity

page 8 Fix table 1

Numbering of the pages and line numbering are confusing, please improve it.

Section 4.3 the sentence started with Elevated ..... should to be rewritten.

Fig.1 part B in my opinion there is a crosstalk between auxin and SL for example in the case of nutrient deficiency. This is cinfirmed for the model plant M.truncatula DOI 10.1007/s11240-018-1393-x

Author Response

Reviewer 1

Dear authors, your review summarized a lot of investigations on Arabidopsis root development in respornse to environmental conditios, hormones and hormonal crosstalk in stress conditions. The data collected in your review will be very helpfull for the researchers working in the field of Arabidopsis root development.

R: The authors appreciate the positive opinion of the reviewer about the contribution of our work to the field and are very grateful for all the valuable corrections and suggestions.

I have few remarks that need to be improved.

Abstarct page 3 line 87 change to has high phenotipic plasticity

R:We have corrected this to “phenotypic plasticity” (now line 21).

page 8 Fix table 1

R: Thank you for this observation, the original format of table 1 was altered because of the Word template, we have corrected it.

Numbering of the pages and line numbering are confusing, please improve it.

R:It has been corrected

Section 4.3 the sentence started with Elevated ..... should to be rewritten.

We have re-written this with an appropriate word (line 487)

Fig.1 part B in my opinion there is a crosstalk between auxin and SL for example in the case of nutrient deficiency. This is cinfirmed for the model plant M.truncatula DOI 10.1007/s11240-018-1393-x

R:Thank you for the observation, it has been described the crosstalk between auxin and SL in plant growth and development, but their relationship under certain stresses is not completely understood (Faizan et al., 2020). In Arabidopsis, we only found that Ruyter-Spira et al., (2011) addressed the relationship between strigolactones and regarding PR, they observe that in response to SL, auxin levels and PINs expression are downregulated; we added this information to our manuscript (line 157).

We know that there is complementary and interesting information in other plants, but our emphasis is in how hormones altered Arabidopsis PR growth as we think our comparative study between different stress conditions will help us to understand deeply hormone function.  Therefore, as a result of this comparative work, we have found that only GA functions as a primary root promoter under different abiotic stress conditions (Table 2); we hope that this review helps the readers to know which information is available for the hormone function on Arabidopsis primary root growth.

Reviewer 2 Report

In the present review authors tried to fulfill an excessively difficult task of analyzing involvement of plant hormones in root growth responses to a wide range of external factors reported for Arabidopsis plants (water deficit, application of exogenous ABA, oxidative stress, toxic ions and availability of mineral nutrients). They based it on analysis of the effects of these factors on hormonal metabolism and transport relating it to the effects of application of exogenous hormones (mainly ABA) and phenotype of Arabidopsis plants with genetically modified hormonal metabolism, transport and perception. The choice of Arabidopsis is understandable, since a lot of Arabidopsis mutants and transgenic plants are available. It is a huge review (it is difficult to count the pages, since its succession is interrupted) overloaded with information, which authors tried to summarize in tables that are not clear enough. An attempt to discuss excessively great amount of information results in contradictions and in that many aspects remain unclear. I propose authors to concentrate on the effects of water shortage resulting from salinity and action of neutral osmotically active substances. I have analyzed this part attentively and my remarks will show that even part of the review demands revision. References should be cited with greater accuracy and some mechanisms should be clearer described. Authors should try to relate the data obtained with Arabidopsis to those of other species. Of course, Arabidopsis is a useful model plant, but the regularities discovered in Arabidopsis would be useless unless they may be applied to other plants species.

I hope that authors will manage with the shortened version of this review with the focus on water shortage and that afterwards they might prepare a readable review addressing effects of other distinct external factors. This will make it easier for readers to follow attentively important information collected by the authors.

My remarks concerning the part of review on the effects of low water availability is arranged according to position of sentences, to which my remarks are related:

  1. Since the review addresses effects on primary root growth, root structure should be described in more details. Such a description (lines 115-117) like: “These cell layers, detected from outside to inside, are lateral root cap (only in the meristematic area), epidermis, cortex, endodermis and pericycle which surround the central vascular tissue” - is unclear and should be better supplied with a figure. If authors mention columella (line 128), they should explain what it is.
  2. Lines 167-173. Opinion that cytokinins merely inhibit root growth is rather popular. Still it may be an oversimplification. Authors should read and cite articles showing that the size and activity of the root apical meristem were markedly reduced in the triple mutant in cytokinin receptors (Higuchi et al., PNAS. 2004. 101. 8821-8826) indicating necessity of cytokinin signaling for proper root growth. Similar results were obtained by Nishimura et al (The Plant Cell, 2004. 16, 1365–1377), who showed that growth and development of the ahk2 ahk3 ahk4 triple mutant were markedly inhibited in various tissues and organs, including the roots. Authors mention these mutants, but only below, when involvement of cytokinins in stress responses is discussed.
  3. (lines 205-206): “Auxin is a promoter while ethylene is a repressor of PR …[67].” This is inaccurate citing. In the cited article of Quin et al it is said that “Auxin and ethylene act synergistically in the regulation of primary root elongation”. This means that both hormones inhibit root elongation. I like the opinion that low concentration of auxins may stimulate root growth, but it should be supported by proper reference.
  4. The role of different hormones on primary root growth under several abiotic 258 stress conditions. … 4. 1 Low water availability. This is the key section of the review and I think a bit more should be told here about importance of root adaptive responses. Authors should emphasize that maintenance of plant water status during stress may be the result of maximized water uptake by increased root growth. Although this mechanism is not likely to act under severe stress, still importance of activation of root growth (at least relative activation in comparison with shoot growth) should be mentioned. Promotive response of water shortage on root growth was frequently detected in field experiments and less frequently – in laboratory. Still activation of arabidopsis root growth under drought has been addressed (e.g., in the article of Durand et al., Plant Physiology. 2016, 170, 1460–1479).
  5. Terms “osmotic” and “salt-stress” should be properly characterized. Authors mentioned, but should state it clearer that osmotic component (low availability of water) is characteristic of both salt stress and action of neutral osmotically active compounds like PEG or manitol. It is the latter influence that is frequently called “osmotic stress”. But to make the difference between them clearer, authors should mention that in this case (unlike salinity) neutral osmotics are meant. To emphasize the difference between salt-stress and action of neutral osmotics, it is important to mention that in the former case plants experience detrimental effects of accumulation of toxic ions alongside with osmotic effects. Authors seemingly tried to mention this by telling that (lines 272-274) “Under salt stress conditions, the PR initial growth reduction is due to the osmotic pressure of the salt and latter is caused by the inability of the roots to contend with the toxic effect of the ions”. However, the sentence is unclear (did not they mean later and not latter?)
  6. Lines 292-294. “ABA inhibition by fluridone, an ABA synthesis inhibitor, generates plants with longer PRs under moderate osmotic stress conditions (−0.37 MPa) compared to WT plants [121]”– These experiments with fluridone suggest that ABA inhibits root elongation in PEG-treated plants, which is opposite to what is depicted in Figure 2 (“osmotic”). Experiments of Sharp et al. (J Exp Bot, 55, 2343–2351) fits better with the figure.
  7. Lines 295-296. “Contrary, fluridone treatment prevents relative PR growth rate under high salt stress conditions” – the results of the cited references are inaccurately described. It is said in the article that “Arabidopsis roots transferred to 140 mM NaCl media supplemented with the ABA biosynthesis inhibitor fluridone exhibited reduced growth rates during the recovery phase of the salt response” and below it is said that “These experiments showed that ABA biosynthesis was most critical during the homeostasis and recovery phases to promote growth, while fluridone treatment during the first 8 h of the salt response had little effect on later growth”. So ABA was more important in these experiments for root growth control during recovery and not salt-stress itself. Anyhow, here again the results are opposite to the scheme in Figure 2, where ABA is shown to inhibit root growth in salt-stressed plants.
  8. (sorry, but lines are not numbered).“ABA is a water-stress plant hormone necessary to withstand low water availability stress responses and ABA accumulation is necessary for PR maintenance under this stress [221].” I like this statement, but do not understand how it accords with the Figure 2, where ABA is shown to inhibit root growth under salt-stress. “Water stress” is rather unclear term (it may be equally applied to flooding). Water deficit or shortage is more precise and it can be applied to salt stress, but then the statement is not correct (see figure 2 about the salt stress and ABA).
  9. Interaction of ABA and ethylene is simplified. They mostly act as antagonist. Mechanism of this is in that ABA may prevent ethylene production by plants (see the article of Sharp et al.), while ethylene may prevent ABA accumulation and signaling. Here and in many other cases authors should read more about cross-talk between hormones.
  10. When discussing cytokinin mutants authors do not distinguish different types of arr mutants. They may refer to Type B cytokinin response regulators which are positive regulators of cytokinin response (like ARR1 and ARR12 mentioned in the tables and text), while Type A ARR (like ARR5 also mentioned in the present review) are negative regulation of cytokinin response. This is important and should be specified.
  11. It is not clear from the text , why BIN2 mutants are less sensitive to ABA exogenous application and BRI1 mutant (bri1-9) is more sensitive to ABA application than WT plants (Figure 2B) [198], while both of them are brassinosteroid insensitive.
  12. Huge table 2 is overloaded with information, which is unclearly presented making it difficult either to read or understand it. It may be divided into several tables.

I have more remarks concerning this part of review, but I do not see much sense in providing them unless authors agree to reduce their review to the analysis of one external factor (water deficit at present with further perspective of submitting reviews on other factors like oxidative stress, availability of mineral nutrients and toxic metals). At any case I cannot endorse publication of the whole review.

Author Response

Reviewer 2

In the present review authors tried to fulfill an excessively difficult task of analyzing involvement of plant hormones in root growth responses to a wide range of external factors reported for Arabidopsis plants (water deficit, application of exogenous ABA, oxidative stress, toxic ions and availability of mineral nutrients). They based it on analysis of the effects of these factors on hormonal metabolism and transport relating it to the effects of application of exogenous hormones (mainly ABA) and phenotype of Arabidopsis plants with genetically modified hormonal metabolism, transport and perception. The choice of Arabidopsis is understandable, since a lot of Arabidopsis mutants and transgenic plants are available. It is a huge review (it is difficult to count the pages, since its succession is interrupted) overloaded with information, which authors tried to summarize in tables that are not clear enough. An attempt to discuss excessively great amount of information results in contradictions and in that many aspects remain unclear. I propose authors to concentrate on the effects of water shortage resulting from salinity and action of neutral osmotically active substances. I have analyzed this part attentively and my remarks will show that even part of the review demands revision. References should be cited with greater accuracy and some mechanisms should be clearer described. Authors should try to relate the data obtained with Arabidopsis to those of other species. Of course, Arabidopsis is a useful model plant, but the regularities discovered in Arabidopsis would be useless unless they may be applied to other plants species.

R: Thank you for all the valuable corrections, suggestions and contributions to improve our work. We have changed table 2 to supplementary table S1 with the aim of facilitating the reading of the review. We have decided to maintain all the sections because it is relevant for us to explain and compare the Arabidopsis root phenotype under contrasting environmental conditions. Using all the information regarding Arabidopsis primary root growth we have seen that the only hormone that is a positive regulator of PR growth in all types of abiotic stresses studied in our review, is GA, while the other hormones respond differently depending on the abiotic stress conditions. The GA result was a surprise for us and we think this review as it is, provides important and interesting information about the specific function of each hormone and its molecular basis on root development and plasticity under specific environmental conditions.

I hope that authors will manage with the shortened version of this review with the focus on water shortage and that afterwards they might prepare a readable review addressing effects of other distinct external factors. This will make it easier for readers to follow attentively important information collected by the authors.

R: Thank you for your suggestion, but that would be another review with a different focus.

My remarks concerning the part of review on the effects of low water availability is arranged according to position of sentences, to which my remarks are related:

  1. Since the review addresses effects on primary root growth, root structure should be described in more details. Such a description (lines 115-117) like: “These cell layers, detected from outside to inside, are lateral root cap (only in the meristematic area), epidermis, cortex, endodermis and pericycle which surround the central vascular tissue” - is unclear and should be better supplied with a figure. If authors mention columella (line 128), they should explain what it is.

R: Since the review is already quite extensive we do not consider necessary to delve into the structure of the root and add another figure, as this can be found in many different articles (Scheres & Wolkenfelt, 1998; Dolan et al., 1993; Luijten et al., 2009 Ivanov and Dubrovsky, 2013). But, we have tried to make a clearer explanation of the root structure and added a brief explanation about the columella cell type (now lines 52-63). 

  1. Lines 167-173. Opinion that cytokinins merely inhibit root growth is rather popular. Still it may be an oversimplification. Authors should read and cite articles showing that the size and activity of the root apical meristem were markedly reduced in the triple mutant in cytokinin receptors (Higuchi et al., PNAS. 2004. 101. 8821-8826) indicating necessity of cytokinin signaling for proper root growth. Similar results were obtained by Nishimura et al (The Plant Cell, 2004. 16, 1365–1377), who showed that growth and development of the ahk2 ahk3 ahk4 triple mutant were markedly inhibited in various tissues and organs, including the roots. Authors mention these mutants, but only below, when involvement of cytokinins in stress responses is discussed.

R: Thank you very much for the observations, we have added the information for the triple mutant of the AHK receptors, underlining its relevance in root development along with the contradictory results we have found. Different alleles of AHK mutants growing under control conditions do not give a precise conclusion of CK effect over PR growth compared to other mutants in CK signaling pathway or biosynthesis that are cited. Besides, many of the mutant phenotypes described in the different articles demonstrate the inhibitory effect of CK in control conditions. But, interestingly, we found the differential response of AHKs and discussed this further in the text related to the controversy of whether CK plays a negative or a positive function in different plant stress responses on primary root growth. For example, in the low water availability section, we found that the AHK mutants respond differently to variations in the osmotic potential (now lines 105-108).

  1. (lines 205-206): “Auxin is a promoter while ethylene is a repressor of PR …[67].” This is inaccurate citing. In the cited article of Quin et al it is said that “Auxin and ethylene act synergistically in the regulation of primary root elongation”. This means that both hormones inhibit root elongation. I like the opinion that low concentration of auxins may stimulate root growth, but it should be supported by proper reference.

R: We have re-written this paragraph to be clearer, although auxins have a biphasic effect, different mutants show that auxins function as a positive regulator of primary root growth. When exogenous ethylene is added to auxin mutants it leads to a reduced sensitivity to ethylene in PR growth, showing crosstalk between these hormones (now lines 139-143)

  1. The role of different hormones on primary root growth under several abiotic 258 stress conditions. … 4. 1 Low water availability. This is the key section of the review and I think a bit more should be told here about importance of root adaptive responses. Authors should emphasize that maintenance of plant water status during stress may be the result of maximized water uptake by increased root growth. Although this mechanism is not likely to act under severe stress, still importance of activation of root growth (at least relative activation in comparison with shoot growth) should be mentioned. Promotive response of water shortage on root growth was frequently detected in field experiments and less frequently – in laboratory. Still activation of arabidopsis root growth under drought has been addressed (e.g., in the article of Durand et al., Plant Physiology. 2016, 170, 1460–1479).

R: Thank you for the suggestion, we have added information regarding the root phenotype and the root-shoot ratio under water deficiency in Arabidopsis (now lines 196-197)

  1. Terms “osmotic” and “salt-stress” should be properly characterized. Authors mentioned, but should state it clearer that osmotic component (low availability of water) is characteristic of both salt stress and action of neutral osmotically active compounds like PEG or manitol. It is the latter influence that is frequently called “osmotic stress”. But to make the difference between them clearer, authors should mention that in this case (unlike salinity) neutral osmotics are meant. To emphasize the difference between salt-stress and action of neutral osmotics, it is important to mention that in the former case plants experience detrimental effects of accumulation of toxic ions alongside with osmotic effects. Authors seemingly tried to mention this by telling that (lines 272-274) “Under salt stress conditions, the PR initial growth reduction is due to the osmotic pressure of the salt and latter is caused by the inability of the roots to contend with the toxic effect of the ions”. However, the sentence is unclear (did not they mean later and not latter?)

R: We have defined the concepts of “osmotic” and “salt-stress” in order to clarify the differences between both types of stresses (lines 199-204). We corrected the sentence in line 205

  1. Lines 292-294. “ABA inhibition by fluridone, an ABA synthesis inhibitor, generates plants with longer PRs under moderate osmotic stress conditions (−0.37 MPa) compared to WT plants [121]”– These experiments with fluridone suggest that ABA inhibits root elongation in PEG-treated plants, which is opposite to what is depicted in Figure 2 (“osmotic”). Experiments of Sharp et al. (J Exp Bot, 55, 2343–2351) fits better with the figure.

R: We fixed figure 2, ABA function negatively in response to osmotic stress and positively during salt stress.

  1. Lines 295-296. “Contrary, fluridone treatment prevents relative PR growth rate under high salt stress conditions” – the results of the cited references are inaccurately described. It is said in the article that “Arabidopsis roots transferred to 140 mM NaCl media supplemented with the ABA biosynthesis inhibitor fluridone exhibited reduced growth rates during the recovery phase of the salt response” and below it is said that “These experiments showed that ABA biosynthesis was most critical during the homeostasis and recovery phases to promote growth, while fluridone treatment during the first 8 h of the salt response had little effect on later growth”. So ABA was more important in these experiments for root growth control during recovery and not salt-stress itself. Anyhow, here again the results are opposite to the scheme in Figure 2, where ABA is shown to inhibit root growth in salt-stressed plants.

R: We have corrected figure 2, ABA function negatively in response to osmotic stress and in a positive way during salt stress. And also fixed it in the text and mentioned that ABA and fluridone in salt stress have an inhibitory effect in PR growth specifically in late phases of the salt stress response, in the called: homeostasis phase (lines 223-225)

  1. (sorry, but lines are not numbered).“ABA is a water-stress plant hormone necessary to withstand low water availability stress responses and ABA accumulation is necessary for PR maintenance under this stress [221].” I like this statement, but do not understand how it accords with the Figure 2, where ABA is shown to inhibit root growth under salt-stress. “Water stress” is rather unclear term (it may be equally applied to flooding). Water deficit or shortage is more precise and it can be applied to salt stress, but then the statement is not correct (see figure 2 about the salt stress and ABA).

R: We have corrected figure 2, ABA function positively during salt stress and we also have changed the sentence to a better understanding regarding low water availability and ABA.

  1. Interaction of ABA and ethylene is simplified. They mostly act as antagonist. Mechanism of this is in that ABA may prevent ethylene production by plants (see the article of Sharp et al.), while ethylene may prevent ABA accumulation and signaling. Here and in many other cases authors should read more about cross-talk between hormones.

R: Thanks for the recommendation, since we restricted our research to Arabidopsis, the articles of Sharp, 2002 and Sharp and LeNoble, 2002 are focused on maize. Reports in Arabidopsis indicates that, as we pointed in the review, ABA positively impacts ethylene signaling to suppress root growth (Sun et al., 2018), ABA induces ethylene biosynthesis leading to the inhibition of primary root growth and ABA responses require normal ethylene signaling (Quin et al., 2019) (lines 344-356).

  1. When discussing cytokinin mutants authors do not distinguish different types of arr mutants. They may refer to Type B cytokinin response regulators which are positive regulators of cytokinin response (like ARR1 and ARR12 mentioned in the tables and text), while Type A ARR (like ARR5 also mentioned in the present review) are negative regulation of cytokinin response. This is important and should be specified.

R: Thank you for your observation, table 1 was modified to indicate the two kinds of ARR genes, and in the text, we specified what type of ARR gene is cited (lines 111,112, 372)

  1. It is not clear from the text , why BIN2 mutants are less sensitive to ABA exogenous application and BRI1 mutant (bri1-9) is more sensitive to ABA application than WT plants (Figure 2B) [198], while both of them are brassinosteroid insensitive.

R: BIN2 is a negative regulator of BR signaling, thus in the triple mutant bin2-3 bil1 bil2 the BR  signal is active, while bin2-1 is a gain of function mutant and is like the BR signal is off (insensitive). BRI1 is the receptor of BR, in which the mutants are insensitive to BR, like bin2-1, and both are hypersensitive to ABA. We added more information to make this statement clearer (lines 380-387).

  1. Huge table 2 is overloaded with information, which is unclearly presented making it difficult either to read or understand it. It may be divided into several tables.

R: We made table 2 since the analysis of mutants gives us huge information related to hormonal response in different stress conditions that was confusing to understand in the text. We will include this table as supplementary material (Table S1) to show all the data that helped us to arrive at our conclusions.

I have more remarks concerning this part of review, but I do not see much sense in providing them unless authors agree to reduce their review to the analysis of one external factor (water deficit at present with further perspective of submitting reviews on other factors like oxidative stress, availability of mineral nutrients and toxic metals). At any case I cannot endorse publication of the whole review.

R: Thank you for all your help and suggestions in the low water availability section. We have revised the whole manuscript and rewrite the sentences that could not be completely clear.

Round 2

Reviewer 2 Report

In the present review authors analyzed involvement of plant hormones in root growth responses to a wide range of external factors reported for Arabidopsis plants (water deficit, application of exogenous ABA, oxidative stress, toxic ions and availability of mineral nutrients). They based it on analysis of the effects of these factors on hormonal metabolism and transport relating it to the effects of application of exogenous hormones (mainly ABA) and phenotype of Arabidopsis plants with genetically modified hormonal metabolism, transport and perception. The choice of Arabidopsis is understandable, since a lot of Arabidopsis mutants and transgenic plants are available. I advised authors to limit their review to some of factors and not all of them, but they insisted on importance of analysis of all the listed affects. Although I planned to refuse from reviewing the MS unless authors shorten it, I now decided to analyze it, since authors really performed a great and valuable work and information collected by them will be useful and interesting for readers. Still authors need to revise the text. They summarized such a great amount of information, that its description is inevitably shortened and not clear enough in many cases. Numerous abbreviated gene names should be deciphered each time, when their involvement is described and not only in the table or when they are first mentioned. The article is not limited in size and reminding readers what the genes control would make it easier and more interesting for readers. I also found several inaccurate citations, which I show below.

Authors have followed the remarks I made during my first revision and in the present one I shall only present my further comments.

  1. Lines 146-147. “Interestingly we have found with this approximation that GA is the only hormone that promote primary root growth in all stress conditions found.” – the review does show that the effects of GA on root growth are really much more universal than those of other hormones. I have objection concerning the word “all stress conditions”. I failed to find references to the effects of GA on root growth under the action of all heavy metals other than Cd.
  2. PINs are mentioned numerous times and nowhere in the text is their function explained. I advise authors to mention that PIN-FORMED (PIN) efflux carriers facilitate the auxin flow out of cells.
  3. In their response to my initial comments authors emphasized importance of GA. In this case they should pay more attention to DELLA, which is the most important component in GA action and although authors mention DELLA in many cases their function is nowhere clearly deciphered. I advise authors to mention that DELLA proteins, are repressors of plant development inhibiting the growth of plant organs through the action on numerous transcription factors and that GA reverses these effects.
  4. Lines 745-746. “in the longitudinal axis of the PR, O2- is accumulated in the MZ, while H2O2 does it in the EZ”. I think it is important to emphasize how this distribution of hydrogen peroxide and super oxide influence PR growth. The phrase from the article cited in the present MS may be useful [188]. It is said that “maintenance of cellular proliferation requires an accumulation of superoxide radical , whereas cellular differentiation requires elevated H2O2 levels”. This will help to explain the opposite effects of these ROS on root growth, since accelerated differentiation implies an arrest of cell division and extension, i.e. inhibition of root growth.
  5. Section concerning involvement of auxins in the responses to oxidative stress conditions (starts from line 735). I think that here it is important to cite references showing effects of auxins on antioxidant system. E.g., it is possible to cite the article of Tiburski et al., Acta Physiol Plant (2009) 31:249–260. Exogenous auxin regulates H2O2 metabolism in roots of tomato (Lycopersicon esculentum Mill.) seedlings affecting the expression and activity of CuZn-superoxide dismutase, catalase, and peroxidase.
  6. The section on hormone crosstalk under oxidative stress conditions describes only interaction between auxin-ABA, while the reference [196] allows discussion of interaction between BR and ethylene.
  7. Lines 890-891. “loss of function mutants on auxin synthesis (taa1-1, yuc9 and the double mutant yuc8 yuc9), auxin signaling (the gain of function solitary root1 (slr-1; IAA14) or the mutants aux1-7 and pin2” - it is easy to notice that unlike other genes the function of AUX1-7 and PIN2 is not deciphered. I think it reasonable to mention that those are mutants in genes coding for auxin carriers (auxin transport mutants).
  8. Lines 972-973. “The application of JA in the loss of function mutant arf7arf19 or in the gain of function mutant slr-1, leads to shorter PR during Al stress, compared with mutant plants exposed to Al only”. – the section is about interaction between JA-auxin. But slr1-1 mutant seemingly has nothing to do with this problem, since it is GA mutant (slender mutant slr1-1 is a recessive mutation resulting in a constitutive gibberellin (GA) response phenotype).
  9. Line 984. I think it is better to decipher what coil-2 mutant is and how it is related to the problem of JA-ethylene interaction.
  10. Lines 1041-1050. It would be good to comment on why the increase in the production of JA is a means to manage with stress, while the synthesis of SA also increases, but resistance is achieved by the decrease in its level.
  11. Lines 1126-1127. “while the double mutant that is hypersensitive to ABA (hab1-1 abi1-2), which encodes protein phosphatases type 2C that are negative regulators of the ABA signaling” – I think this is unclear sentence. Hab-1 means Hypersensitive to ABA1 while abi1 is 2ABA-Insensitive.
  12. Lines 1329-1330“Furthermore, lowering concentration, from 50mM to 1mM of KNO3 induces an increment in auxin content in roots and PR growth reduction [236].” – the cited article is about lateral and not primary roots.
  13. Lines 1332-1333. “Also, the PR growth of afb3-1 mutant is not inhibited by nitrate (5 1333 mM) under hydroponic condition as observed in WT plants” – it would be wise to remind that this is auxin receptor mutant.
  14. Lines 1334-1335. “In this case, the nitrate exogenous application (5mM) does not have the same promoting activity as reported in other works [232,239]” – this is unclear citing. Nothing in said about this mutant in [232], while [239] is not about the effects of nitrate, but about the action of glutamate
  15. Line 1355. “the PR length is greater under high N”. This sounds strange, since high nitrates are known to inhibit PR growth (see, e.g., the work of Walch-Liu et al.,2006 (Annals of Botany)). Actually, concentration of nitrates was not high, but optimal in the article cited by the authors of the present review (1.5 mM). It was just compared with very low N concentration.
  16. Since both NRT1 and NPF nitrate transporters are mentioned, it may be worth mentioning that they belong to the same family (NRT1/PTR FAMILY (NPF)).
  17. Line “nahG mutant “ – this is not mutant but transgenic line over expressing NAHG.
  18. Lines 1781-1782. “ACC (0.1 μM to 10 μM) co-treatment with high or low Pi concentrations inhibits PR growth [245].” I think that in this respect the article cited under the number of 257 may be more useful. It is said there that inhibiting ethylene production (with aminoethoxyvinyl-glycine) or action (with 1-methylcyclopropene) increased elongation in high phosphorus and decreased it in low phosphorus.
  19. Lines 1809-1810. “The negative regulation of PR growth by OPR3 under Pi deficiency is dependent on GA accumulation” _ I find it worth mentioning here that OPR3 is 12-oxophytodienoate reductase involved in the biosynthesis of jasmonic acid (JA)
  20. Lines 1868-1869. “Under K-starved conditions the PR length is reduced, which correlates with a decrease in CK levels in order to cope with K deficiency [266]. Unexpectedly, the OE of IPT3, reduces PR growth under K-deficient and the loss of function ipt1,3,5,7 is insensitive to K deficiency [266]” – This does not fit with information that “CK-deficient ipt1,3,5,7 mutant enhanced root growth … under low K conditions”. See the cited reference [266].
  21. Line 1952“Low Fe suppresses PR growth inhibition of BRZ-treated roots” – It is worth mentioning that Brassinazole (Brz) is a specific brassinosteroid biosynthesis inhibitor.
  22. Lines 1985-1987. “Auxin has a dual role on PR growth as PIN2 enhances it whereas AUX1 inhibits it during B deficiency. “ – this contradiction may be clarified, if authors mention that PIN and AUX1 auxin transporter perform opposite roles: while PIN carries auxins out of the cells., while AUX1 facilitates their uptake.

Author Response

Reviewer 2.

In the present review authors analyzed involvement of plant hormones in root growth responses to a wide range of external factors reported for Arabidopsis plants (water deficit, application of exogenous ABA, oxidative stress, toxic ions and availability of mineral nutrients). They based it on analysis of the effects of these factors on hormonal metabolism and transport relating it to the effects of application of exogenous hormones (mainly ABA) and phenotype of Arabidopsis plants with genetically modified hormonal metabolism, transport and perception. The choice of Arabidopsis is understandable, since a lot of Arabidopsis mutants and transgenic plants are available. I advised authors to limit their review to some of factors and not all of them, but they insisted on importance of analysis of all the listed affects. Although I planned to refuse from reviewing the MS unless authors shorten it, I now decided to analyze it, since authors really performed a great and valuable work and information collected by them will be useful and interesting for readers. Still authors need to revise the text. They summarized such a great amount of information, that its description is inevitably shortened and not clear enough in many cases. Numerous abbreviated gene names should be deciphered each time, when their involvement is described and not only in the table or when they are first mentioned. The article is not limited in size and reminding readers what the genes control would make it easier and more interesting for readers. I also found several inaccurate citations, which I show below.

R: The authors are very grateful for the evaluation and for the valuable corrections and suggestions. We have added the meaning of abbreviations the first time that these are mentioned in the text and we included a brief explanation on the function of each gene.

Authors have followed the remarks I made during my first revision and in the present one I shall only present my further comments.

  1. Lines 146-147. “Interestingly we have found with this approximation that GA is the only hormone that promote primary root growth in all stress conditions found.” – the review does show that the effects of GA on root growth are really much more universal than those of other hormones. I have objection concerning the word “all stress conditions”. I failed to find references to the effects of GA on root growth under the action of all heavy metals other than Cd.

R: Thank you for the observation, we have changed the sentence to: “all stress conditions described in this review” (line 80-81).

  1. PINs are mentioned numerous times and nowhere in the text is their function explained. I advise authors to mention that PIN-FORMED (PIN) efflux carriers facilitate the auxin flow out of cells.

R: We have added this information the first time the PINs are mentioned in the text (line 97-103).

  1. In their response to my initial comments authors emphasized importance of GA. In this case they should pay more attention to DELLA, which is the most important component in GA action and although authors mention DELLA in many cases their function is nowhere clearly deciphered. I advise authors to mention that DELLA proteins, are repressors of plant development inhibiting the growth of plant organs through the action on numerous transcription factors and that GA reverses these effects.

R: Thank you very much for the suggestion, we have added this information the first time the DELLAs proteins are cited (line 130-132).

  1. Lines 745-746. “in the longitudinal axis of the PR, O2- is accumulated in the MZ, while H2O2 does it in the EZ”. I think it is important to emphasize how this distribution of hydrogen peroxide and super oxide influence PR growth. The phrase from the article cited in the present MS may be useful [188]. It is said that “maintenance of cellular proliferation requires an accumulation of superoxide radical, whereas cellular differentiation requires elevated H2O2 levels”. This will help to explain the opposite effects of these ROS on root growth, since accelerated differentiation implies an arrest of cell division and extension, i.e. inhibition of root growth.

R: Thank you for the recommendation, we have incorporated this new information in the text (line 459-461)

  1. Section concerning involvement of auxins in the responses to oxidative stress conditions (starts from line 735). I think that here it is important to cite references showing effects of auxins on antioxidant system. E.g., it is possible to cite the article of Tiburski et al., Acta Physiol Plant (2009) 31:249–260. Exogenous auxin regulates H2O2 metabolism in roots of tomato (Lycopersicon esculentum Mill.) seedlings affecting the expression and activity of CuZn-superoxide dismutase, catalase, and peroxidase.

R: Thank you for the suggestion, but we will not add the tomato information as we are trying to avoid information from other plants and concentrate only on what has been done in Arabidopsis. However, we found that tir afb2 double mutant positively regulates the activity of the antioxidant enzyme ASCORBATE PEROXIDASE 1 (APX1); accordingly, this mutant has less H2O2 levels in response to Methyl Viologen (Iglesias et al., 2010) (line 482-484).

  1. The section on hormone crosstalk under oxidative stress conditions describes only interaction between auxin-ABA, while the reference [196] allows discussion of interaction between BR and ethylene.

R: Thank you for the observation, we have added the crosstalk between BR- ethylene (line 520-524)

  1. Lines 890-891. “loss of function mutants on auxin synthesis (taa1-1, yuc9 and the double mutant yuc8 yuc9), auxin signaling (the gain of function solitary root1 (slr-1; IAA14) or the mutants aux1-7 and pin2” - it is easy to notice that unlike other genes the function of AUX1-7 and PIN2 is not deciphered. I think it reasonable to mention that those are mutants in genes coding for auxin carriers (auxin transport mutants).

R: We have added this information (line 565) and along the manuscript.

  1. Lines 972-973. “The application of JA in the loss of function mutant arf7arf19 or in the gain of function mutant slr-1, leads to shorter PR during Al stress, compared with mutant plants exposed to Al only”. – the section is about interaction between JA-auxin. But slr1-1 mutant seemingly has nothing to do with this problem, since it is GA mutant (slender mutant slr1-1 is a recessive mutation resulting in a constitutive gibberellin (GA) response phenotype).

R: The name of this mutant refers to solitary root 1 (slr-1), which is IAA14. We have added its IAA number (slr-1/iaa14) to be clearer (line 607 and 612).

  1. Line 984. I think it is better to decipher what coil-2 mutant is and how it is related to the problem of JA-ethylene interaction.

R: Thank you for the suggestion, the mutant was written incorrectly, the correct name is coi1-2, we added that this mutant refers to the JA receptor and included a brief explanation for the JA-ethylene crosstalk (lines 624, 627-628)

  1. Lines 1041-1050. It would be good to comment on why the increase in the production of JA is a means to manage with stress, while the synthesis of SA also increases, but resistance is achieved by the decrease in its level.

R: Thank you for the recommendation, this information is very interesting. We have included new information in each section to explain what the functions of JA and SA are during Cd stress. The positive function of JA against Cd stress is mediated through the expression of genes that encode antioxidant enzymes and the suppression of genes that promote the Cd uptake and its translocation. Whereas, the nahG transgenic plants, with less SA content, display less ROS levels and enhance the level of the antioxidant glutathione (lines 669-671, 677-680, 682-685)

  1. Lines 1126-1127. “while the double mutant that is hypersensitive to ABA (hab1-1 abi1-2), which encodes protein phosphatases type 2C that are negative regulators of the ABA signaling” – I think this is unclear sentence. Hab-1 means Hypersensitive to ABA1 while abi1 is 2ABA-Insensitive.

R: Thank you for the observation. The meaning of the genes HAB and ABI1 is on lines 383 and 247. We have added that both are mutants of protein phosphatases type 2C and are negative regulators of the ABA signaling (line 730). Regarding to ABI1, the abi1-1 mutant, that was first characterized, is a gain of function mutant that is insensitive to ABA, while  abi1-2 is the recessive loss-of-function mutant that is hypersensitive to ABA treatment, showing that ABI1 acts as a negative regulator of ABA signaling (Saez et al., 2006).  

  1. Lines 1329-1330“Furthermore, lowering concentration, from 50mM to 1mM of KNO3 induces an increment in auxin content in roots and PR growth reduction [236].” – the cited article is about lateral and not primary roots.

R: The article concentrates mainly in lateral roots; however, the authors also describe the effect of nitrate availability on primary root growth, as you can verify in figure 1A. We decided not to include this information further in our revision, since, although we see an augment in the DR5 signal in the primary root between 1mM and 50mM of KNO3 treatment, the authors indicate no apparent changes in the DR5 signal of the primary roots under these treatments.

  1. Lines 1332-1333. “Also, the PR growth of afb3-1 mutant is not inhibited by nitrate (5 1333 mM) under hydroponic condition as observed in WT plants” – it would be wise to remind that this is auxin receptor mutant.

R: Thank you for the recommendation, we have added this information (line 796)

  1. Lines 1334-1335. “In this case, the nitrate exogenous application (5mM) does not have the same promoting activity as reported in other works [232,239]” – this is unclear citing. Nothing in said about this mutant in [232], while [239] is not about the effects of nitrate, but about the action of glutamate

R: Thank you for the observation. We have changed this information to the general paragraph about nitrate (lines 785-787) as in both manuscripts (Naulin et al., 2020 (232) and Walch-Liu (239)) the nitrate exogenous application was on primary root growth in WT plants. In addition, Walch-Liu et al 2008 [239] also shows the effect of nitrate (5 mM) on primary root growth (black bar; Figure 1b).

  1. Line 1355. “the PR length is greater under high N”. This sounds strange, since high nitrates are known to inhibit PR growth (see, e.g., the work of Walch-Liu et al.,2006 (Annals of Botany)). Actually, concentration of nitrates was not high, but optimal in the article cited by the authors of the present review (1.5 mM). It was just compared with very low N concentration.

R: Ok, we meant “the effect of the co-treatment of GA3 and high N, resulting in a longer primary root than control conditions without the hormone treatment” and corrected this sentence to be clearer (lines 816-818). We added in the manuscript that high N levels refers to 2500 µM N while low levels to 50 µM  N (Conesa et al., 2020).

  1. Since both NRT1 and NPF nitrate transporters are mentioned, it may be worth mentioning that they belong to the same family (NRT1/PTR FAMILY (NPF)).

R: Thank you for the suggestion. We included that AtNPF6.3/NRT1.1 (that are the same transporter) and NPF3 belong to the NRT1/PTR family (lines 795,819).

  1. Line “nahG mutant “ – this is not mutant but transgenic line over expressing NAHG

R: Thanks for the correction, we have made it in the manuscript (lines 674-676,847).

  1. Lines 1781-1782. “ACC (0.1 μM to 10 μM) co-treatment with high or low Pi concentrations inhibits PR growth [245].” I think that in this respect the article cited under the number of 257 may be more useful. It is said there that inhibiting ethylene production (with aminoethoxyvinyl-glycine) or action (with 1-methylcyclopropene) increased elongation in high phosphorus and decreased it in low phosphorus.

R: Thank you for the observation. However, we think it is useful to leave the ACC information as is consistent with the phenotype observed in two mutants (eto1 and ctr1) with high ethylene response (lines 912-913). In addition, the different phenotypes observed with the inhibitors, is not sufficient to explain the phenotypes observed with ACC. Moreover, it was already written that the ethylene inhibitors could have another function that seems to be partially independent of ethylene (lines 909-910).

  1. Lines 1809-1810. “The negative regulation of PR growth by OPR3 under Pi deficiency is dependent on GA accumulation” _ I find it worth mentioning here that OPR3 is 12-oxophytodienoate reductase involved in the biosynthesis of jasmonic acid (JA)

R: This information was already written a few lines before (line 921): “12-OXOPHYTODIENOATE REDUCTASE 3 (OPR3), a gene responsible for JA biosynthesis”.

  1. Lines 1868-1869. “Under K-starved conditions the PR length is reduced, which correlates with a decrease in CK levels in order to cope with K deficiency [266]. Unexpectedly, the OE of IPT3, reduces PR growth under K-deficient and the loss of function ipt1,3,5,7 is insensitive to K deficiency [266]” – This does not fit with information that “CK-deficient ipt1,3,5,7 mutant enhanced root growth … under low K conditions”. See the cited reference [266].

R: Thank you for this observation. However, the paragraph cited “CK-deficient ipt1,3,5,7 mutant enhanced root growth … under low K conditions” refers to lateral root growth.

As can be seen in Figure 2A (Nam et al., 2012 or 266, (now 273)), the primary root growth of the ipt1,3,5,7 mutant, does not change either under K-sufficient or K-deficient conditions for 7 days compared to WT plants.

  1. Line 1952“Low Fe suppresses PR growth inhibition of BRZ-treated roots” – It is worth mentioning that Brassinazole (Brz) is a specific brassinosteroid biosynthesis inhibitor.

R: Ok, we add this information all over the manuscript (lines 651-652, 895, 978, 1014)

  1. Lines 1985-1987. “Auxin has a dual role on PR growth as PIN2 enhances it whereas AUX1 inhibits it during B deficiency. “ – this contradiction may be clarified, if authors mention that PIN and AUX1 auxin transporter perform opposite roles: while PIN carries auxins out of the cells., while AUX1 facilitates their uptake.

R: Thank you for the recommendation, we have included this explanation all over the text (lines 1045-1046)
